# LDADAM: ADAPTIVE OPTIMIZATION FROM LOW-DIMENSIONAL GRADIENT STATISTICS

**Thomas Robert**[1*], **Mher Safaryan**[2], **Ionut-Vlad Modoranu**[2], **Dan Alistarh**[2]
[1]Institut Polytechnique de Paris (IPP)
[2]Institute of Science and Technology Austria (ISTA)

## ABSTRACT

We introduce LDAdam, a memory-efficient optimizer for training large models, that performs adaptive optimization steps within lower dimensional subspaces, while consistently exploring the full parameter space during training. This strategy keeps the optimizer's memory footprint to a fraction of the model size. LDAdam relies on a new projection-aware update rule for the optimizer states that allows for transitioning between subspaces, i.e., estimation of the statistics of the projected gradients. To mitigate the errors due to low-rank projection, LDAdam integrates a new generalized error feedback mechanism, which explicitly accounts for both gradient and optimizer state compression. We prove the convergence of LDAdam under standard assumptions, and show that LDAdam allows for accurate and efficient fine-tuning and pre-training of language models. Code is available at https://github.com/IST-DASLab/LDAdam.

## 1 INTRODUCTION

Scaling up deep neural networks leads to ever-improving models (Kaplan et al., 2020), but also to ever-increasing memory and compute requirements. In this work, we focus on the memory cost of *optimizer states*, i.e., the additional first- and second-order gradient statistics required during the adaptive optimization process, which in the case of the baseline algorithm Adam (Kingma & Ba, 2017; Loshchilov & Hutter, 2019) reaches up to twice the size of the model. Our goal is to replicate Adam's performance and theoretical guarantees while significantly reducing its memory footprint.

We start from the observation that the optimization landscape of deep networks exhibits low intrinsic dimensionality (Li et al., 2018). Low-rank learning has been successfully leveraged for model *fine-tuning*, with the LoRA technique (Hu et al., 2021) and its many extensions (Dettmers et al., 2023; Liu et al., 2024; Nikdan et al., 2024) becoming standard. However, such adapter-based methods do not extend to end-to-end accurate training (Lialin et al., 2023; Zhao et al., 2024), nor even to fine-tuning on more demanding tasks (Zhang et al., 2023; Nikdan et al., 2024; Liu et al., 2024). Despite promising results such as GaLore (Zhao et al., 2024), designing memory-efficient full-parameter optimizers which leverage low-dimensional gradient statistics with provable convergence and good practical performance remains an open challenge.

**Contributions.** We propose *Low-Dimensional Adam (LDAdam)*, a memory-efficient adaptive optimizer which, for the first time, incorporates low-rank compression of *both gradients and optimizer states*, with convergence guarantees under standard assumptions. The goal of the algorithm is to perform optimization steps within subspaces of lower dimension, while introducing mechanisms to 1) maintain relevant information from the last seen subspaces and 2) recover information lost due to low-rank compression. To reach this goal, our work is based on three main techniques. First, we formulate projection-aware update rules that allow us to perform adaptive optimization on dynamically changing coordinate systems and subspaces. Second, we leverage *block power iteration* (Bentbib & Kanber, 2015) at each step to efficiently and accurately project the gradient and optimizer states. Third, we use a generalized error feedback mechanism for both gradients and optimizer states. Moreover, we prove the convergence of LDAdam under standard assumptions and provide empirical evidence supporting its training efficiency.

---

*Work performed as an intern at ISTA. Correspondence to dan.alistarh@ist.ac.at.

When performing adaptive optimization *within low-dimensional subspaces*, a key challenge is to adapt the optimizer states to changes in the projection map. Adapting gradient statistics to new subspaces is not obvious and comes with two challenges. First, standard adaptive optimization algorithms are anisotropic: that is, the parameter-wise learning rate tuning implies that the choice of coordinate system used to express the gradient matrix impacts the optimization process. Second, the second-order statistics are nonlinear, which complicates estimation of the gradient statistics and error compensation. We address the first challenge by carefully selecting the coordinate system used for low-rank compression, i.e. the basis given by the most significant singular vectors. The second challenge is addressed by approximating the gradient statistics in the current subspace given the raw moments estimates in the previous subspace. Thus, with LDAdam, optimizer step is performed from the projection-aware intermediate optimizer states and the projected gradient, rather than from the optimizer states at the previous iteration and the full-rank gradient (Kingma & Ba, 2017). In turn, this leads to stronger convergence guarantees, and better practical performance.

The choice of projection map at each step is key to low-rank compression. Performing singular value decomposition (SVD) at each step would add prohibitive overhead; instead, we use *block power iteration with warm initialization* (Bentbib & Kanber, 2015; Vogels et al., 2020), which can achieve a tight approximation of the most significant singular vectors of successive gradients, at low cost. To preserve the information from the last iterations kept within the optimizer states, we show that one can perform the block power iteration on a barycenter of the newly computed gradient and an average of the previous gradients. Finally, to mitigate the errors due to low-rank projection, we introduce a new variant of the error feedback mechanism (Seide et al., 2014; Karimireddy et al., 2019; Alistarh et al., 2018) that supports compression of *both gradient and optimizer states*. Moreover, we propose a memory efficient implementation of it by reusing the space allocated to gradient accumulation.

The resulting LDAdam method is memory-efficient: for a weight layer of shape $n \times m$, the space cost of optimizer states is $nr + 2rm$, where $r$ is the projection rank, while the cost of Adam's state is $2nm$. This matches the efficiency of the popular GaLore algorithm (Zhao et al., 2024), the closest prior work in this area; and can bring major gains in practice, as usually $r \ll \min(m, n)$. Relative to GaLore, LDAdam has the advantages of tracking gradient evolution in the adapted subspace at each step, and correcting for the projection error. This leads to better practical performance, but also to theoretical guarantees of convergence under standard assumptions. Specifically, for smooth non-convex objectives, LDAdam can preserve the asymptotic convergence rate of AMSGrad (Reddi et al., 2019), the provably-convergent version of Adam (see Theorem 1). Moreover, we show even faster rates for objectives that obey the Polyak-Łojasiewicz condition (see Theorem 2).

We validate the practical performance of LDAdam via an efficient PyTorch implementation. We apply LDAdam for *fine-tuning* RoBERTa (Liu et al., 2019) and Llama-family (Touvron et al., 2023) models on the GLUE (Wang et al., 2018) and Grade-School Math (GSM) (Cobbe et al., 2021) benchmarks, respectively. In addition, we provide *pre-training* results for Llama-type models with 130M and 350M parameters. Across all tasks and models, LDAdam shows competitive accuracy relative to Adam, but with much smaller optimizer states relative to the baseline, and comparable runtime. Relative to GaLore, LDAdam shows consistently higher accuracy, for matching memory costs and similar runtime. Ablations show that this is a consequence of both our projection-aware intermediate updates and error correction.

In summary, we show that it is possible to achieve both theoretical guarantees and strong practical performance when performing adaptive optimization from low-dimensional gradient statistics. Our techniques may also be relevant to areas such as distributed optimization, while our implementation is relevant in reducing the practical memory overheads of training large language models (LLMs).

## 2  RELATED WORK

**Memory-Efficient First-Order Methods in Optimization.** Gradient statistics are used to determine the descent direction (Rumelhart et al., 1986) and to adapt the learning rate per parameter (Duchi et al., 2011; Hinton, 2012). Adam (Kingma & Ba, 2017) combines both ideas in a bias-corrected design, and AdamW (Loshchilov & Hutter, 2019) adds weight decay. Although it is the de facto optimizer for deep neural networks, Adam/AdamW has been the subject of several extensions aimed at reducing its memory footprint. For example, Adafactor (Shazeer & Stern, 2018), CAME (Luo et al., 2023) and Adam-mini (Zhang et al., 2024a) rely on a factorized representation of the second-order statistic to reduce memory. Quantization has also been proposed to reduce the

footprint of optimizer states (Dettmers et al., 2022), and fusing computation of the backward pass and the optimizer step (Lv et al., 2024b;a) has also proven effective.

**Error Feedback in Distributed Optimization.** Several methods have been proposed to reduce the communication overhead in distributed optimization: quantization (Alistarh et al., 2017), sparsification (Alistarh et al., 2018; Lin et al., 2020) and low-rank approximation (Vogels et al., 2020). Error feedback (Seide et al., 2014) has been shown to be effective in improving the empirical performance of biased gradient compression methods (Karimireddy et al., 2019; Vogels et al., 2020) methods and key to ensuring their convergence (Stich et al., 2018; Alistarh et al., 2018). MicroAdam (Modoranu et al., 2024) only stores sparse gradients from each optimization step, and uses error feedback for correction, but reconstructs the full optimizer state at each step, instead of compressing it.

**Parameter-Efficient Fine-Tuning (PEFT).** A natural approach for reducing memory overheads is to train less parameters (Han et al., 2024). This idea led to additive PEFT, where new trainable modules are added to the model architecture (Houlsby et al., 2019; Li & Liang, 2021); selective PEFT, where only certain parameters of the original model are retrained (Guo et al., 2021); and reparametrized PEFT, where new modules are trained and merged with the original parameters (Hu et al., 2021; Nikdan et al., 2024; Liu et al., 2024). However, methods that train significantly less parameters tend to underperform for pre-training and fine-tuning on challenging tasks.

**Optimization via Low-Rank Gradient Projection.** GaLore (Zhao et al., 2024), which partly motivated our work, performs adaptive optimization in a low-dimensional subspace that is updated periodically. The adaptive optimizer is fed with a low-rank representation of the gradients obtained via SVD, thus saving memory, and produces a low-rank representation of the descent direction, which is then up-projected to update the model parameters. The idea is similar to PEFT in that low-rank gradient projection yields selective training in the coordinate system induced by SVD. The addition of low-rank components to achieve full-rank training is also proposed in ReLoRA (Lialin et al., 2023) or COLA (Xia et al., 2024). However, unlike PEFT methods, GaLore computes and exploits the full gradient of the original model. The final algorithm allows pre-training of 7B parameter language models on consumer GPUs. However, it can only do so with a very small batch size and without the ability to accumulate gradients over multiple mini-batches.

**GaLore Comparison.** Relative to GaLore, we propose the following new techniques: (1) we introduce a new projection-aware update rule (see Sections 3.1 and 3.2), which avoids accumulating low-rank gradients from *different* subspaces into the optimizer states, and performs frequent updates; (2) to correct for projection errors in *both* gradients and optimizer states, we introduce a new generalized error feedback mechanism (see Section 3.4). The convergence of GaLore is only guaranteed under a very strong "stable-rank" assumption, whereas we show convergence under standard conditions. Experimentally, we show that these improvements lead to consistently improved accuracy, and that they can be implemented efficiently in terms of both memory and runtime.

Several orthogonal improvements such as quantization (Dettmers et al., 2022; Zhang et al., 2024b), per-layer weight update (Lv et al., 2024b), and weight factorization (Jaiswal et al., 2024) have been proposed to further improve projection-based methods. Many of these improvements are compatible with LDAdam, and we plan to investigate them in future work.

## 3 THE LOW-DIMENSIONAL ADAM (LDADAM) ALGORITHM

### 3.1 THE NEED FOR A PROJECTION-AWARE UPDATE RULE

**Standard Adaptive Optimization.** We denote $f$ the loss function to minimize, $\theta_t$ the model parameters at step $t$, $G_t = \nabla_\theta f(\theta_t) \in \mathbb{R}^{n \times m}$ the gradient at $\theta_t$, $g_t$ its mini-batch stochastic counterpart, $\eta_t$ the learning rate, and $\epsilon$ a positive scalar used for numerical stability. The standard adaptive optimization relies on the gradient statistics estimates $m_t$ and $v_t$, which are obtained from the exponential moving average of the gradients at rate $\beta_1$ and the squared gradients at rate $\beta_2$, respectively. With $\hat{m}_t = \frac{m_t}{1-\beta_1^t}$ and $\hat{v}_t = \frac{v_t}{1-\beta_2^t}$ their unbiased counterpart, Adam's update is: $\theta_{t+1} = \theta_t - \eta_t \frac{\hat{m}_t}{\sqrt{\hat{v}_t}+\epsilon}$.

**The Challenges Behind Low-Rank Updates.** When adapting the learning subspaces to capture the low-rank gradient structure, one expects the optimizer states to retain information from previous iterations. However, frequent updates of the low-rank projection causes significant challenges: first, projections are lossy and gradient information is lost; second, since the projections may be different

between steps, subspace adaptation may alter the representation of the low-dimensional optimizer states. To illustrate this second point, we briefly discuss the GaLore solution (Zhao et al., 2024).

Let $r$ be the compression rank, and $\mathcal{U}_t^\top$ and $\mathcal{U}_t$ be the projection and back-projection matrices we use at step $t$. $\mathcal{U}_t^\top \in \mathbb{R}^{r \times n}$ yields a truncation to its first $r$ rows of the gradient matrix written in an adapted coordinate system, leading to a low-dimensional representation of the gradient with shape $r \times m$. Multiplication with the matrix $\mathcal{U}_t$ is then used to express the compressed gradient in the canonical high-dimensional coordinate system. From now on, let $m_t$ and $v_t$ (resp. $\hat{m}_t$ and $\hat{v}_t$) be the first and second moment estimates (resp. unbiased estimates) in the *low-dimensional space* induced at step $t$ by $\mathcal{U}_t^\top$, and let $M_t = \mathcal{U}_t \cdot m_t$ denote the back-projected low-rank counterpart of $m_t$.

In this context, Equation 1 describes the GaLore dynamics (Zhao et al., 2024, Algorithm 2):

$$\theta_{t+1} = \theta_t - \eta_t \boldsymbol{\mathcal{U}_t} \cdot \frac{\hat{m}_t}{\sqrt{\hat{v}_t}+\epsilon} = \theta_t - \frac{\eta_t}{\sqrt{\hat{v}_t}+\epsilon} \frac{1-\beta_1}{1-\beta_1^t} \sum_{\tau=1}^{t} \beta_1^{t-\tau} \boldsymbol{\mathcal{U}_t} \cdot \boldsymbol{\mathcal{U}_\tau^\top} \cdot g_\tau. \tag{1}$$

Notice that, in the above equation, the change in the low-dimensional projection basis between steps is ignored: as can be seen in the right-hand sum, gradients are projected and back-projected using *different maps*, i.e. $\boldsymbol{\mathcal{U}_t}$ and $\boldsymbol{\mathcal{U}_\tau^\top}$, respectively. Moreover, the statistics of gradients projected into different subspaces are accumulated on *the same momentum buffers*. The issue we raise here (highlighted in bold in Equation 1) is partially mitigated in practice by heuristics: occasional updates of the projection map and exponential decay rates lead to only a fraction of the steps being effectively resulting from accumulation of compressed gradients in different subspaces. However, we will see experimentally that this leads to a drop in accuracy relative to the "correct" projections.

## 3.2 THE LDADAM UPDATE RULE

**Overview.** With LDAdam, we address two key challenges in low-rank updates: we introduce a projection-aware update rule that accounts for the change in the low-dimensional representation of the optimizer states, and we provide a generalized error feedback mechanism that accounts for the loss of gradient accuracy due to low-rank compression. Compared to standard adaptive optimization step, optimizer states are replaced by low-dimensional intermediate optimizer states, denoted by $m_{t-1/2}$, $v_{t-1/2}$; and the gradient is replaced by the accumulator $a_t = \mathcal{U}_t^\top \cdot A_t$ obtained by low-rank projection of the high-dimensional accumulator $A_t = g_t + \xi_t$, where $\xi_t$ is the (generalized) error feedback buffer, storing projection errors. In the following, we build the LDAdam algorithm step-by-step. The end-to-end construction is provided in Algorithm 1.

Our first objective is to find a mechanism to transfer gradient information from one lower dimensional vector space to another. For this, we take inspiration from orthogonal projections, which provide the best approximation in terms of $\ell_2$ error. The projection-aware update rule for the first moment estimate (Algorithm 1 Line 8) follows from the multiplication with the matrix $\mathcal{U}_t^\top \cdot \mathcal{U}_{t-1}$, which stands for both the projection matrix and the change of basis matrix. The matrix $\mathcal{U}_t^\top \cdot \mathcal{U}_{t-1} \in \mathbb{R}^{r \times r}$ is low-dimensional and thus allows efficient transition between subspaces, as no intermediate high-dimensional vector is created. We obtain the following update:

$$m_t = \beta_1 \text{PROJ}_{\mathcal{U}_t}(M_{t-1}) + (1-\beta_1)a_t = \beta_1 \underbrace{\mathcal{U}_t^\top \cdot \mathcal{U}_{t-1} \cdot m_{t-1}}_{m_{t-1/2}} + (1-\beta_1)a_t. \tag{2}$$

**Optimizer States Compression.** Note already that projection yields compression. Yet, the discrepancy between successive subspaces leads to a loss of information contained in the optimizer states. We address this by enforcing slight shifts between subspaces, and by generalizing error feedback to account for the updating of optimizer states. The information lost on gradient momentum due to compression, and that needs to be reintroduced via the error feedback mechanism, is given by:

$$M_{t-1} - \mathcal{U}_t \mathcal{U}_t^\top \cdot M_{t-1} = \mathcal{U}_{t-1} \cdot m_{t-1} - \mathcal{U}_t \cdot m_{t-1/2}. \tag{3}$$

**Estimating the Statistics of Projected Gradients.** Although reliable, the orthogonal projection of the optimizer state does not apply to the estimation of the second-order statistic. In particular, Adam does not rely solely on linear operations and is anisotropic (see Appendix A). To overcome these challenges, we interpret Adam's optimizer states as statistical estimates of the first two moments of each gradient coordinate. Indeed, Adam's optimization step $t$ holds for an approximation of $(\mathbb{E}_{t,\beta_1}[G^{e_i}]/\sqrt{\mathbb{E}_{t,\beta_2}[(G^{e_i})^2]})_{i \leq n}$, where $\mathcal{E}_t = (e_1, \dots, e_n)$ is the basis provided by the canonical

parameterization of the model, $G^{e_i} = \langle G, e_i \rangle$ a column of the gradient matrix, and $\mathbb{E}_{t,\beta}[\cdot]$ is the exponential time-weighted expectation from time $t$ with decay-rate $\beta$.

We rely on this observation to rewrite Adam's update in any given coordinate system $\tilde{\mathcal{E}}_{t+1}$: we build $\hat{m}_{t+1/2}$ and $\hat{v}_{t+1/2}$ as statistical estimates of $(\mathbb{E}_{t,\beta_1}[G^{\tilde{e}_i}])_{i \leq n}$ and $(\mathbb{E}_{t,\beta_2}[(G^{\tilde{e}_i})^2])_{i \leq n}$. It follows from $G^{\tilde{e}_i} = \sum_{j \leq n} \langle \tilde{e}_i, e_j \rangle G^{e_j}$ that the matrix of change of basis $(\langle \tilde{e}_i, e_j \rangle)_{i,j \leq n}$ provides transition between coordinate systems. Its truncation to $i, j \leq r$ induces a rank-$r$ orthogonal projection of the gradients, and thus holds for the transition between subspaces:

$$\mathbb{E}_{t,\beta_1}[G^{\tilde{e}_i}] = \sum_{j=1}^n \langle \tilde{e}_i, e_j \rangle \mathbb{E}_{t,\beta_1}[G^{e_j}] \approx \sum_{j=1}^r \langle \tilde{e}_i, e_j \rangle (\hat{m}_t)_j = (\mathcal{U}_{t+1}^\top \cdot \mathcal{U}_t \cdot \hat{m}_t)_i. \qquad (4)$$

Linearity of expectation leads to an autoregressive update rule for the first moment estimate (see Equation 4). With $(\langle \tilde{e}_i, e_j \rangle)_{i,j} = (\mathcal{U}_{t+1}^\top \cdot \mathcal{U}_t)_{i,j}$ the estimation of the first-order statistic in the coordinate system $\tilde{\mathcal{E}}_{t+1}$ resolves to an orthogonal projection when considering only the first $r$ components (i.e., after truncation to $i, j \leq r$), and the above discussion on compression of optimizer states remains valid. Following our statistical approach, we obtain the projection-aware update rule for the second moment estimates (Algorithm 1 Line 9) from the approximation below:

$$\mathbb{E}_{t,\beta_2}[(G^{\tilde{e}_i})^2] = \sum_{j=1}^n \langle \tilde{e}_i, e_j \rangle^2 \mathbb{E}_{t,\beta_2}[(G^{e_j})^2] + \sum_{k \neq l}^n \langle \tilde{e}_i, e_k \rangle \langle \tilde{e}_i, e_l \rangle \mathbb{E}_{t,\beta_2}[G^{\tilde{e}_k} G^{\tilde{e}_l}]$$
$$\approx \sum_{j=1}^r \langle \tilde{e}_i, e_j \rangle^2 (\hat{v}_t)_j + \sum_{k \neq l}^r \langle \tilde{e}_i, e_k \rangle \langle \tilde{e}_i, e_l \rangle (\hat{m}_t)_k (\hat{m}_t)_l. \qquad (5)$$

The approximation of the second-order statistics in a new coordinate system requires the estimation of the covariance between the gradient coordinates (see Equation 5). Although too large to store, the covariance can be approximated by the product of first-order moment estimates, assuming independence between gradient coordinates. The latter assumption is enforced on average over the gradient matrix columns by the nature of the coordinate system inherited from SVD. In practice, we guarantee a positive estimate of the second-order statistics by clipping value to zero if necessary.

### 3.3 LEARNING SUBSPACE ADAPTATION

LDAdam's memory savings stem from the low-dimensional structure of the moments estimate. The compression strategy, defined by a choice of truncated coordinate systems, sets the exploration of the parameter space, and its design meets two criteria: (i) the learning subspace must be adapted at each step to integrate error feedback; (ii) the compression operator must be set to approximate the current gradient and to preserve the information stored in the optimizer states.

**Projection Map Computation.** Although singular value decomposition (SVD) provides an optimal low-rank approximation, its computational cost makes it prohibitive in our case. We follow PowerSGD (Vogels et al., 2020) to efficiently approximate the most significant singular vectors: we perform a single block power iteration (Bentbib & Kanber, 2015) initialized with the approximation from the previous optimization step; and apply the Gram-Schmidt process to derive the projection map (Algorithm 1 Line 6). In addition, we propose to start the whole process by computing SVD of the first gradient.

**Learning A Smooth Subspace Transition.** LDAdam relies at each step on compression of the gradient and the optimizer states via a unique low-rank projection map. Instead of fitting two projection maps and interpolate between them, we propose to fit a single projection map to the interpolation of the gradient and an average of the previous gradients (Algorithm 1 Line 5). We introduce an interpolation factor $\rho$ and perform a block power iteration to approximate SVD of the matrix: $B_t = \rho\, \mathcal{U}_{t-1} \cdot \hat{m}_{t-1} + (1 - \rho) A_t$. The interpolation factor helps to balance the accuracy of the gradient compression with the preservation of the optimizer states. We suggest setting $\rho = \beta_1$, i.e. computing the optimal subspace for gradient momentum compression.

### 3.4 THE GENERALIZED ERROR FEEDBACK MECHANISM

Error Feedback was introduced for distributed optimization (Seide et al., 2014) to account for gradient compression errors by reintroducing them into the next iteration gradient. We extend this mechanism to account for loss of information on both the gradient and the optimizer states. We keep the same structure of the error feedback mechanism: the optimizer is fed by a unique accumulator that is a sum of the gradient and the error buffer.

**Unbiased Error Buffer Loading.** With gradient and optimizer states decayed at different rates within the optimizer step, we must adjust the error buffer loading strategy of the optimizer states projection error by the factor $\beta_1/(1 - \beta_1)$ to recover the first moment estimate. Furthermore, the condition under which the error buffer loading strategy is consistent with the second moment estimate is given by: $\beta_2 = (1 - \beta_2)(\frac{\beta_1}{1-\beta_1})^2$. We suggest setting $\beta_2 = 0.99$ and $\beta_1 \approx 0.908$, as the slight decrease in the decay rate of the second moment estimate helps to adapt faster to a new learning subspace. Equation 6 describes the loading strategy of the generalized error feedback mechanism (Algorithm 1 Line 16).

$$\xi_{t+1} = [A_t - \mathcal{U}_t \cdot a_t] + \tfrac{\beta_1(\beta_2)}{1-\beta_1(\beta_2)}[\mathcal{U}_{t-1} \cdot m_{t-1} - \mathcal{U}_t \cdot m_{t-1/2}]. \tag{6}$$

**Implementing Error Feedback.** A naive implementation of the error feedback mechanism would add memory overhead equal to the model size. Since the gradient and the error buffer are added together before the optimization step is performed, we store the error buffer in the variable used for gradient accumulation (Algorithm 1 Lines 16 and 3)(e.g. in PyTorch, modify the `optimizer.zero_grad` function to store the error buffer in the `p.grad` variable). This results in an error feedback mechanism that does not require any further memory for optimizer states.

---

**Algorithm 1** LDAdam ($\diamond$ Practical View Only, $g_t \in \mathbb{R}^{n \times m}$ / $\diamond$ Analytical View Only, $g_t \in \mathbb{R}^d$ )

---

    **Hyperparameters:** step size $\eta_t$; decay rates $\beta_1$, $\beta_2$
    **LDAdam Hyperparameters:** projection rank $r$; interpolation factor $\rho$
    **LDAdam Hyperparameters:** $\diamond$ contraction factor $q_r \in [0, 1)$; $\rho = \beta_1$
1:  **Initialization:** $m_0 = 0$; $v_0 = 0$; $A_0 = 0$, $\diamond \mathcal{U}_0 = \mathrm{SVD}(g_0)$; $\diamond \tilde{v}_0 = 0$; $\xi_1 = 0$
2:  **for** $t = \{1, 2, \ldots, T\}$ **do**
     **Gradient accumulation and error buffer unloading**
3:       $\diamond A_t = A_t + g_t$
4:       $\diamond A_t = \xi_t + g_t$
     **Learning subspace adaptation**
5:       $B_t = \rho \, \mathcal{U}_{t-1} \hat{m}_{t-1} + (1 - \rho) A_t$
6:       $\diamond \mathcal{U}_t = \mathrm{GRAM\text{-}SCHMIDT}(B_t B_t^\top \cdot \mathcal{U}_{t-1})$
7:       $\diamond \mathcal{U}_t$ is any $d \times r$ orthogonal matrix such that $\|(I - \mathcal{U}_t \mathcal{U}_t^\top) B_t\| \le q_r \|B_t\|$
     **Optimizer states projection-aware update**
8:       $m_{t-1/2} = \mathcal{U}_t^\top \mathcal{U}_{t-1} m_{t-1}$
9:       $v_{t-1/2} = (1 - \beta_2^{t-1}) \left| (\mathcal{U}_t^\top \mathcal{U}_{t-1})^2 \cdot (\hat{v}_{t-1} - (\hat{m}_{t-1})^2) + (\mathcal{U}_t^\top \mathcal{U}_{t-1} \cdot \hat{m}_{t-1})^2 \right|$
     **Optimizer states Adam-type update**
10:     $a_t = \mathcal{U}_t^\top A_t$
11:     $m_t = \beta_1 m_{t-1/2} + (1 - \beta_1) a_t$
12:     $v_t = \beta_2 v_{t-1/2} + (1 - \beta_2) a_t^2$
13:     $\diamond \tilde{v}_t = \max(v_t, \|\tilde{v}_{t-1}\|_{\max})$
     **Model update**
14:     $\diamond \theta_{t+1} = \theta_t - \eta_t \mathcal{U}_t \cdot \frac{\hat{m}_t}{\sqrt{\hat{v}_t} + \epsilon}$
15:     $\diamond \theta_{t+1} = \theta_t - \eta_t \mathcal{U}_t \cdot \frac{m_t}{\sqrt{\tilde{v}_t} + \epsilon}$
     **Error buffer loading**
16:     $A_{t+1} = (A_t - \mathcal{U}_t \cdot a_t) + \frac{\beta_1}{1-\beta_1}(\mathcal{U}_{t-1} \cdot m_{t-1} - \mathcal{U}_t \cdot m_{t-1/2})$ $\diamond = \xi_{t+1}$
17: **end for**

---

## 4   CONVERGENCE GUARANTEES FOR LDADAM

**Analytical Overview.** We now present our theoretical convergence results for LDAdam. We first provide/discuss an "analytical" perspective on our approach in Algorithm 1. Then we introduce and discuss the analytical assumptions used in the theory, along with two theoretical convergence rates.

The outlined steps closely resemble those in the practical view of the algorithm. One notable difference is the introduction of a new AMSGrad-type normalization, $\tilde{v}_t = \max(v_t, \|\tilde{v}_{t-1}\|_{\max})$ in line 13. This represents uniform version of the original AMSGrad technique (Reddi et al., 2019), $\tilde{v}_t = \max(v_t, \tilde{v}_{t-1})$, which enforces coordinate-wise monotonicity for the diagonal preconditioning. In our setup, as we transition between different low-dimensional spaces, the preconditioning matrix

is no longer diagonal, and scaling factors may change after such a transition. To ensure that adaptive preconditioning remains monotonic, we enforce a monotonic spectrum of the preconditioning.

To avoid unnecessary complications in the notations and derivations, in this section we assume that the stochastic gradients $g_t$ and other variables derived from the gradients, such as momentum, are *vectors*. We remind the reader that, in practice and in our implementation, gradients are viewed/represented as 2D matrices. In our analysis, vector gradients should be treated as the columns of the gradient matrix. Nevertheless, all steps in Algorithm 1 are still applicable when dealing with matrix gradients. Specifically, given a matrix gradient $g_t$, we obtain a matrix $B_t$, for which we assume that, regardless of the low-rank compression method used (e.g., SVD or power iterations), the contractive inequality $\|(I - \mathcal{U}_t^\top \mathcal{U}_t) B_t\|_{\mathrm{F}} \leq q_r \|B_t\|_{\mathrm{F}}$ holds with respect to the Frobenious norm. We also incorporate the debiasing factor $\sqrt{1 - \beta_2^t}/(1 - \beta_1^t)$ in $\eta_t$.

**Convergence Guarantees for General Smooth Non-convex Functions.** We first outline the analytical assumptions under which we establish LDAdam's convergence guarantees.

**Assumption 1** (Lower bound and smoothness). *The loss function $f \colon \mathbb{R}^d \to \mathbb{R}$ is L-smooth and lower bounded by some $f^* \in \mathbb{R}$: $\|\nabla f(\theta) - \nabla f(\theta')\| \leq L\|\theta - \theta'\|, \quad$ for any $\theta, \theta' \in \mathbb{R}^d$.*

**Assumption 2** (Unbiased and bounded stochastic gradient). *For all iterates $t \geq 1$, the stochastic gradient $g_t$ at $\theta_t$ is unbiased and uniformly bounded by some constant $G \geq 0$:*

$$\mathbb{E}[g_t] = \nabla f(\theta_t), \quad \|g_t\| \leq G.$$

**Assumption 3** (Bounded variance). *For all iterates $t \geq 1$, the variance of the stochastic gradient $g_t$ at $\theta_t$ is uniformly bounded by some constant $\sigma^2 \geq 0$: $\mathbb{E}[\|g_t - \nabla f(\theta_t)\|^2] \leq \sigma^2$.*

All three assumptions, including the bounded gradient condition, are standard and commonly used in adaptive optimization literature (Reddi et al., 2019; Chen et al., 2019; Défossez et al., 2022; Modoranu et al., 2024). Attempting to relax the bounded gradient condition, some works (Shi et al., 2020; Zhang et al., 2022; Wang et al., 2024) resorted to using a strong growth condition (which is not necessarily a relaxation over bounded gradient condition) or bounded stochastic noise conditions (Li et al., 2023; Hong & Lin, 2024). Taniguchi et al. (2024) achieved an optimal rate by modifying Adam and only slightly relaxing the bounded gradient assumption by requiring the expected gradient norm to be bounded. The bounded gradient condition in Assumption 2 directly implies a bounded variance condition with a constant $\sigma^2 = 2G^2$. Therefore, for the purpose of asymptotic analysis, one can omit Assumption 3. However, we will show that in the non-convex convergence rate, the $\sigma^2$-term decays at a rate of $1/\sqrt{T}$, while all other $G$-terms decay at a faster rate of $1/T$. Theorem 1 states our general non-convex convergence result.

**Theorem 1** (**Non-convex convergence rate**). *Let Assumptions 1, 2 and 3 hold. Then, choosing step-size $\eta = \min(\frac{\epsilon}{4LC_0\sqrt{1+C_2}}, \frac{1}{\sqrt{T}})$, LDAdam (Algorithm 1) satisfies*

$$\frac{1}{T} \sum_{t=1}^{T} \mathbb{E}[\|\nabla f(\theta_t)\|^2] \leq \frac{2C_0}{\sqrt{T}} \left( f(\theta_1) - f^* + \frac{L\sigma^2}{\epsilon} \right) + \mathcal{O}(\frac{G^3}{T})$$

*with constants $C_0 := \sqrt{\frac{1+\beta_2}{1-\beta_2} \frac{(1-\beta_1(1-q_r))^2}{(1-\beta_1)^2(1-q_r)^2} G^2 + \epsilon}$ and $C_2 = \frac{\beta_1 + (1-\beta_1)q_r^2}{(1-\beta_1)^2(1-q_r)^2}$.*

**Discussion.** Compared to the standard stochastic gradient descent (Ghadimi & Lan, 2016), the leading term $1/\sqrt{T}$ of the obtained rate recovers the optimal non-convex convergence speed. Additionally, the asymptotic rate $\mathcal{O}(\frac{1+\sigma^2}{\sqrt{T}} + \frac{1}{T})$ aligns with the rate of uncompressed AMSGrad in the stochastic non-convex setup (Zhou et al., 2024). Therefore, the introduced low-rank compression framework along with the error feedback mechanism does not hurt the asymptotic convergence with respect to $T$. The slowdown caused by compression in the leading term is only through the constant $C_0 = \mathcal{O}(\frac{1}{1-q_r})$, where the factor $1 - q_r \in (0, 1]$ measures the aggressiveness of the compression. Note that our theory applies to any low-rank compression with $r \geq 1$. More compression corresponds to a smaller rank $r$ for the projections and a smaller factor $1 - q_r$. For instance, when using low-rank compression via SVD decomposition, one can prove $1 - q_r = \Theta(\frac{r}{d})$ for any rank $r \geq 1$. In this case, the leading term of the rate becomes $\mathcal{O}(\frac{d}{r} \frac{1}{\sqrt{T}})$, indicating that the convergence slows down in the worst case by a factor of $d/r$ due to compression. The proof can be found in Appendix E.2.

**Convergence Rate for Non-Convex Functions under the PL Condition.** Next, we show faster convergence for LDAdam when the loss function satisfies the PL condition.

**Assumption 4** (PL-condition). *For some constant $\mu > 0$ the loss function $f$ satisfies the Polyak-Łojasiewicz (PL) inequality:* $\|\nabla f(\theta)\|^2 \geq 2\mu(f(\theta) - f^*)$, *for any $\theta \in \mathbb{R}^d$.*

**Theorem 2. (PL convergence rate)** *Let Assumptions 1, 2, 3 and 4 hold. Then, choosing step-size $\eta = \min(\eta_0, \frac{2C_0 \log T}{\mu T})$, LDAdam (Algorithm 1) satisfies*

$$\mathbb{E}[f(\theta_{T+1})] - f^* \leq \frac{\log T}{T}\left(\frac{2LC_0^2\sigma^2}{\mu^2\epsilon} + \frac{6C_0(1+C_1)G^2}{\mu\sqrt{\epsilon}}\right) + \widetilde{\mathcal{O}}\left(\frac{G^4}{T^2}\right)$$

*with constants $C_1 := \frac{\beta_1 + (1-\beta_1)q_r}{(1-\beta_1)(1-q_r)}$ and $\eta_0 := \min(\frac{\epsilon}{16LC_0}, \frac{C_0(1-\beta_1)(1-q_r)}{2\mu}, \frac{\epsilon^{3/4}}{6L\sqrt{C_0C_2}})$.*

**Discussion.** Notably, the leading term of the convergence rate under the PL condition improves to $\mathcal{O}(\frac{\log T}{T})$, compared to $\mathcal{O}(\frac{1}{\sqrt{T}})$ in the general non-convex setting. Faster rates of $\widetilde{\mathcal{O}}(\frac{1}{T})$ for adaptive optimization algorithms under the PL condition have been established when $\beta_2 \to 1$ (He et al., 2023) or through the use of the AMSGrad trick (Modoranu et al., 2024). In LDAdam, we apply a uniform version of AMSGrad normalization with arbitrary $\beta_2 < 1$ and derive the rate for compression with any rank $r \geq 1$. Hence, similar to the general non-convex case, we show the best-known convergence rate in the leading term, up to a logarithmic factor. While the last term has higher-order constant dependencies, these are negligible due to the $T^2$ damping effect. Thus, the theory suggests that the algorithm's convergence rate remains comparable to that of the uncompressed version, with only a constant factor affected by the compression rank. To quantify the slowdown caused by compression, we again consider low-rank compression via SVD decomposition, as discussed earlier. As noted before, $C_0 = \mathcal{O}(\frac{1}{1-q_r}) = \mathcal{O}(\frac{d}{r})$, and similarly, from the expression of $C_1$, we have $C_1 = \mathcal{O}(\frac{1}{1-q_r}) = \mathcal{O}(\frac{d}{r})$. Therefore, the leading term of the rate becomes $\mathcal{O}((\frac{d}{r})^2 \frac{\log T}{T})$, with a compression slowdown factor of $(\frac{d}{r})^2$. The detailed proof is deferred to Appendix E.3.

## 5 TRAINING MEMORY FOOTPRINT AND RUNTIME

LDAdam's memory savings over Adam (Kingma & Ba, 2017) (see Table 1 for comparison) follow from the low dimensionality of the optimizer states: for gradients of shape $n \times m$ with $n \leq m$, the gradient statistics in a learning subspace of rank $r$ are stored as tensors of shape $r \times m$, and the additional projection matrix is of shape $n \times r$. As a result, the memory footprint of LDAdam's optimizer states is only a fraction of the size of the model. For an $n \times n$ layer, the compression ratio of LDAdam relative to Adam is of $\frac{3}{2}\frac{r}{n}$. Table 6 and Figure 3 reports peak memory for fine-tuning and pre-training tasks.

With the error buffer stored in the gradient variable, the error feedback mechanism is memory neutral. However, this comes at the cost of not supporting gradient clipping and per-layer weight updates (Lv et al., 2024b; Zhao et al., 2024). The latter allows memory savings proportional to the model size, but at the expense of gradient accumulation. To be precise, the standard GaLore algorithm does not support gradient accumulation (see Appendix D.2 for a detailed explanation), while its newer "retaining grad" version (Zhao et al., 2024) does, but at the expense of a *memory overhead equal to the size of the model*. Although applicable to most optimizers, per-layer weight updates is not widely adopted, due to the large batch size used for large language model training (Touvron et al., 2023) and the memory-intensive nature of activations in attention layers.

Compared to GaLore, LDAdam adds optimizer states projection-aware update and the generalized error feedback mechanism. Yet, the practical runtime differences are not substantial: on Llama-350M pre-training, LDAdam has a 10% longer runtime compared to GaLore, and is 15% slower than Adam. Table 7 and Figure 3 report runtimes, and Table 5 reports theoretical complexity in terms of standard matrix operations.

## 6 EXPERIMENTS

We empirically evaluate LDAdam for fine-tuning and pre-training large language models against baseline Adam (Kingma & Ba, 2017) and the memory-efficient GaLore (Zhao et al., 2024) (Algorithm 2). Similar to most parameter-efficient fine-tuning (PEFT) methods and to GaLore, we apply low-rank compression only to the two-dimensional matrices of the self-attention layers, e.g. wide embedding and output layers are trained using standard Adam optimizer. For a fair comparison, in

Table 1: Optimizer comparison: parameter count during training for a weight layer of shape $n \times m$ with $n \leq m$ (i.e., left projection), training capabilities, and estimates of optimizer states memory footprint in half precision (models architecture is described in Table 9).

| | Adam | LDAdam | GaLore (retaining grad) | GaLore |
|---|---|---|---|---|
| **Token count** | | | | |
| Weights | $nm$ | $nm$ | $nm$ | $nm$ |
| Gradients | $nm$ | $nm$ | $nm$ | |
| Optimizer States | $2nm$ | $nr + 2rm$ | $nr + 2rm$ | $nr + 2rm$ |
| Gradient Clipping | ✓ | ✗ | ✓ | ✗ |
| Gradient Accumulation | ✓ | ✓ | ✓ | ✗ |
| **Memory estimates** | | | | |
| RoBERTa-base (`r=8`) | 0.46 GB | 0.15 GB | 0.15 GB | 0.15 GB |
| Llama 350M (`r=256`) | 1.37 GB | 0.95 GB | 0.95 GB | 0.95 GB |
| Llama-2 7B (`r=32`) | 25.1 GB | 1.22 GB | 1.22 GB | 1.22 GB |
| Llama-2 7B (`r=512`) | 25.1 GB | 4.87 GB | 4.87 GB | 4.87 GB |

all experiments we do not apply additional layer-wise learning rate scaling, and apply hyperparameters suggested in the papers introducing the algorithm. To ensure reproducibility, the full list of hyperparameters we used is provided in the Appendix C.

**Fine-tuning on GLUE.** We evaluate LDAdam for fine-tuning RoBERTa-base model (Liu et al., 2019) on the General Language Understanding Evaluation (GLUE) benchmark (Wang et al., 2018). RoBERTa-base is a 125M parameters large language model, built upon the encoder only attention-based BERT architecture (Devlin et al., 2019), and pre-trained for natural language understanding (NLU). GLUE benchmark includes a wide variety of NLU tasks with training samples of various size. We mimic memory-constrained setups by training for only 3 epochs with a batch size of 16, and used rank `r=8` compression for LDAdam and GaLore. For a fair comparison, we tune the learning rate over the set {1e-5, 2e-5, ..., 5e-5}. Table 2 reports the best average over 3 seeds (Table 8 reports standard deviation), and Figure 5 shows that the error buffer norm is of the same order of magnitude as the gradient norm, and that their ratio is stable throughout training.

Table 2: Results of fine-tuning RoBERTa-base model on the GLUE benchmark.

| | MRPC | STS-B | CoLA | SST-2 | QNLI | QQP | MNLI | Avg |
|---|---|---|---|---|---|---|---|---|
| Training Samples | 3.7k | 7k | 8.5k | 67k | 105k | 364k | 393k | |
| Adam | 88.97 | 90.08 | 58.57 | 94.34 | 92.81 | 91.38 | 87.85 | 86.28 |
| LDAdam | 88.40 | 90.11 | 59.91 | 95.00 | 92.87 | 91.28 | 87.81 | 86.48 |
| LDAdam no-EF | 88.00 | 89.88 | 56.86 | 95.00 | 92.32 | 89.75 | 86.99 | 85.54 |
| GaLore | 86.19 | 88.97 | 55.12 | 94.15 | 92.01 | 89.86 | 86.80 | 84.73 |

**Fine-tuning on GSM8K.** We evaluate LDAdam for fine-tuning Llama-2 7B model (Touvron et al., 2023) on the GSM8K mathematical reasoning dataset (Cobbe et al., 2021) containing grade school math word problems. We run experiments for ranks `r=32` and `r=512`, on 3 epochs, with a batch size of 32, and tune the learning rate over {5e-5, 6e-5, ..., 9e-5, 1e-4, 2e-4, ..., 5e-4}. Table 3 reports the best average accuracy. We also provide peak memory usage, but also the fixed memory cost of storing model, gradient and activations.

Table 3: Results of fine-tuning Llama-2 7B on the GSM8K dataset.

| | Adam | LDAdam | | LDAdam no-EF | | GaLore | |
|---|---|---|---|---|---|---|---|
| | | `r=32` | `r=512` | `r=32` | `r=512` | `r=32` | `r=512` |
| **Accuracy** | 34.72 | 32.53 | 35.86 | 30.70 | 35.78 | 27.07 | 35.18 |
| **Peak Memory** | 55.34 GB | 32.08 GB | 35.58 GB | 32.08 GB | 35.58 GB | 32.52 GB | 35.58 GB |

Across all fine-tuning experiments, LDAdam outperforms GaLore, and is either on par or better than Adam. When ablating the LDAdam techniques, we observe that LDAdam without error feedback also outperforms GaLore, supporting LDAdam's continuous subspace adaptation strategy over GaLore's sequential update approach. The fact that we can match Adam once error feedback is enabled suggests that this technique is key.

**Pre-training on C4.** We evaluate LDAdam for pre-training Llama models (Touvron et al., 2023) on the C4 dataset (Raffel et al., 2023). Due to limited compute resources, we pre-train smaller 130M, 350M and 1.3B parameter variants (Lialin et al., 2023; Zhang & Sennrich, 2019; Shazeer, 2020) of the Llama-2 models. C4 is a clean version of Common Crawl's web crawl corpus. We adjust the number of training steps according to the Chinchilla scaling law (Hoffmann et al., 2022), i.e. 20 training tokens per model parameter; and use batch size 512 and sequence length 256, accounting to 131 072 tokens per step, thus using the same settings as GaLore. For a fair comparison, we tune the learning rate over the set {5e-4, 1e-3, 5e-3}, and report the best results for ranks `r=8` and `r=256` in Table 4. Figure 1 depicts the pre-training dynamics.

Table 4: Results (validation perplexity ↓) of pre-training Llama models on the C4 dataset.

|  | Adam | LDAdam | | LDAdam no-EF | | GaLore | |
|---|---|---|---|---|---|---|---|
|  |  | `r=8` | `r=256` | `r=8` | `r=256` | `r=8` | `r=256` |
| **Llama 130M** | 24.64 | 23.82 | 22.65 | 45.17 | 25.32 | 67.78 | 26.04 |
| **Llama 350M** | 18.08 | 18.37 | 17.30 | - | 19.85 | - | 20.03 |

Furthermore, we scale the experiments up to the Llama 1.3B parameter model, using the same hyperparameters but fixing the learning rate at $5e-4$ due to limited computational resources. Adam achieves a validation perplexity of 14.86 , while LDAdam with rank `r=16` achieves 14.09.

Figure 1: Pre-training dynamics for Llama 350M (left) and Llama 1.3B (right) on the C4 dataset.

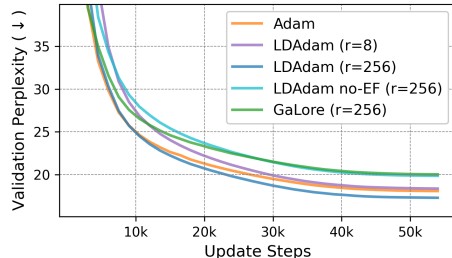 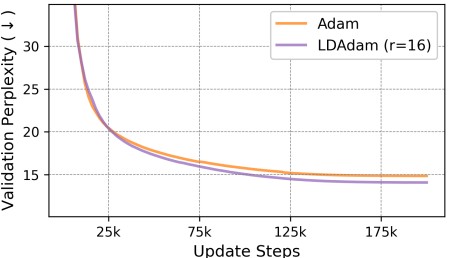

Pre-training experiments also show that LDAdam has comparable performance to Adam, both in terms of convergence and validation accuracy, and confirm the improvements over GaLore. Further, for very high compression rates (i.e., `r=8`), LDAdam still converges, but its no-EF variant does not, nor does GaLore. This highlights the need for an error correction mechanism for pre-training tasks, which are known to require models updates of high rank. For lower compression (i.e. `r=256`), LDAdam outperforms the no-EF version and GaLore, which is correlated to their consistent projection errors. For more details on the impact of the rank on training, see Figure 4. LDAdam even appears to slightly outperform Adam. We attribute this small improvement to compression, as sparse updates might enhance implicit regularization (Zhang et al., 2017; Neyshabur, 2017).

## 7 DISCUSSION

We proposed a new low-rank learning method with the memory-efficient LDAdam optimizer, analyzed its convergence under standard assumptions, and provided empirical evidence of its ability to match the performance of Adam at a fraction of its memory cost. LDAdam relies on Adam to estimate gradient statistics, but its development required the design of a specific intermediate optimizer state update rule. Therefore, efficient implementations of Adam are not directly applicable to LDAdam, and extending our work to other optimizers requires further work. On the experimental side, a natural next step would be to execute large-scale billion-parameter *pre-training* experiments.

# 8 ETHICS STATEMENT

This paper presents work that aims to advance the field of machine learning. We believe that memory-efficient optimization is a step toward democratizing large-scale model training, and thus provides opportunities to foster both the development of new applications and the research in the field. There are many societal concerns about the rapidly growing use of artificial intelligence, we feel that none of them is specifically related to our work.

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

## A  CHALLENGES FOR ESTIMATING THE SECOND-ORDER STATISTIC OF PROJECTED GRADIENTS

**Adam Does Not Rely Solely on Linear Operations.** Adam follows the proposition from AdaGrad (Duchi et al., 2011) to use diagonal approximation to estimate: $((1-\beta_2)\sum_{\tau=0}^{t}\beta_2^{t-\tau}g_\tau \cdot g_\tau^T)^{-\frac{1}{2}}$. This leads to $v_t^{Adam} = (1-\beta_2)\sum_{\tau=0}^{t}\beta_2^{t-\tau}g_\tau \odot g_\tau$ and to the matrix inverse square root being replaced by coordinate wise inverse square root operations. Such an approximation saves both memory and computation, but at the cost of leaving the linear framework. Therefore, one cannot expect to draw an autoregressive linear algebra formula to update the second moment estimate.

**Adam is Anisotropic.** Adaptive optimization learning rate tuning for each parameter is equivalent to a direction-wise rescaling of the training space. It implies that the choice of coordinate system affects the optimization steps. Let $\bar{\mathcal{U}}$ be a fixed orthogonal matrix which acts as rotation in the parameter space and consider the following two minimization problems:

$$(P_I) : \arg\min_{\theta\in\mathbb{R}^{n\times m}} f(\theta) \tag{7}$$

$$(P_{\bar{\mathcal{U}}}) : \arg\min_{\tilde{\theta}\in\mathbb{R}^{n\times m}} f(\bar{\mathcal{U}} \cdot \tilde{\theta}) \tag{8}$$

These two problems are equivalent up to a change of parametrization. However, applying Adam to each of them will lead to optimization process that are not equivalent: adaptive estimate of the second-order statistics of the gradient leads to $\theta_t \neq \bar{\mathcal{U}} \cdot \tilde{\theta}_t$ :

$$\theta_{t+1} = \theta_t - \eta_t \frac{\sqrt{1-\beta_2^t}}{1-\beta_1^t} \frac{(1-\beta_1)\sum_{\tau=1}^{t}\beta_1^{t-\tau}\nabla f(\theta_t)}{\sqrt{(1-\beta_2)\sum_{\tau=1}^{t}\beta_2^{t-\tau}(\nabla f(\theta_t))^2}+\epsilon} \approx \theta_t - \eta_t \frac{\mathbb{E}_{t,\beta_1}[\nabla f(\theta_t)]}{\sqrt{\mathbb{E}_{t,\beta_2}[(\nabla f(\theta_t))^2]}+\epsilon} \tag{9}$$

$$\tilde{\theta}_{t+1} = \tilde{\theta}_t - \eta_t \frac{\sqrt{1-\beta_2^t}}{1-\beta_1^t} \frac{(1-\beta_1)\sum_{\tau=1}^{t}\beta_1^{t-\tau}\bar{\mathcal{U}}^\top\nabla f(\bar{\mathcal{U}}\cdot\tilde{\theta}_t)}{\sqrt{(1-\beta_2)\sum_{\tau=1}^{t}\beta_2^{t-\tau}(\bar{\mathcal{U}}^\top\nabla f(\bar{\mathcal{U}}\cdot\tilde{\theta}_t))^2}+\epsilon} \approx \tilde{\theta}_t - \eta_t \frac{\mathbb{E}_{t,\beta_1}[\bar{\mathcal{U}}^\top\nabla f(\bar{\mathcal{U}}\cdot\tilde{\theta}_t)]}{\sqrt{\mathbb{E}_{t,\beta_2}[(\bar{\mathcal{U}}^\top\nabla f(\bar{\mathcal{U}}\cdot\tilde{\theta}_t))^2]}+\epsilon} \tag{10}$$

Low-rank adaptive optimization follows from $(P_{\mathcal{U}})$ where $\mathcal{U} \in \mathbb{R}^{n\times r}$ is the truncation of the rotation matrix $\bar{\mathcal{U}}$ to its first $r$ rows, and thus yields reparametrization and projection to a lower-dimensional space. However, this leads to $\tilde{\theta} \in \mathbb{R}^{r\times m}$ and the optimization set is restricted to a low-rank subspace. To enable full-parameter training, GaLore's strategy is to keep the model parameter high-rank while performing low-dimensional updates, leading to:

$$\theta_{t+1}^{\text{GALORE}} = \theta_t^{\text{GALORE}} - \mathcal{U}\cdot\left(\eta_t \frac{\sqrt{1-\beta_2^t}}{1-\beta_1^t} \frac{(1-\beta_1)\sum_{\tau=1}^{t}\beta_1^{t-\tau}\mathcal{U}^\top\nabla f(\theta_t^{\text{GALORE}})}{\sqrt{(1-\beta_2)\sum_{\tau=1}^{t}\beta_2^{t-\tau}(\mathcal{U}^\top\nabla f(\theta_t^{\text{GALORE}}))^2}+\epsilon}\right) = \theta_t^{\text{GALORE}} - \mathcal{U}\cdot\left(\eta_t \frac{\hat{m}_t}{\sqrt{\hat{v}_t}+\epsilon}\right) \tag{11}$$

With the projection map $\mathcal{U} = \mathcal{U}_t$ occasionally updated, GaLore's low-rank gradient statistic estimate $\hat{m}_t$ (resp. $\hat{v}_t$) fails to approximate $\mathbb{E}_{t,\beta_1}[\mathcal{U}_t^\top\nabla f(\theta_t^{\text{GALORE}})]$ (resp. $\mathbb{E}_{t,\beta_2}[(\mathcal{U}_t^\top\nabla f(\theta_t^{\text{GALORE}}))^2]$), because gradients projected into different subspaces are accumulated on the same momentum buffers. A simple idea would be to fix a coordinate system for the momentum buffers. However, this would have to be high-dimensional and thus would prevent memory savings. The purpose of LDAdam's projection-aware update rule is to adapt the optimizer states to the new learning subspace and coordinate system, leading to $\hat{m}_{t-1/2} \approx \mathbb{E}_{t-1,\beta_1}[\mathcal{U}_t^\top\nabla f(\theta_{t-1}^{\text{LDADAM}})]$, and $\hat{v}_{t-1/2} \approx \mathbb{E}_{t-1,\beta_2}[(\mathcal{U}_t^\top\nabla f(\theta_{t-1}^{\text{LDADAM}}))^2]$ and enabling gradient accumulation within low-dimensional momentum buffers. One can thus rewrite LDAdam's heuristic as performing the following update:

$$\theta_{t+1}^{\text{LDADAM}} \approx \theta_t^{\text{LDADAM}} - \mathcal{U}_t \cdot \left(\eta_t \frac{\mathbb{E}_{t,\beta_1}[\mathcal{U}_t^\top\nabla f(\theta_t^{\text{LDADAM}})]}{\sqrt{\mathbb{E}_{t,\beta_2}[(\mathcal{U}_t^\top\nabla f(\theta_t^{\text{LDADAM}}))^2]}+\epsilon}\right). \tag{12}$$

As a result, while Adam is anisotropic, LDAdam is isotropic since it doesn't rely on any specific coordinate system but instead always uses the one induced by the SVD of its gradient.

## B  ADDITIONAL MEASUREMENTS

### B.1  TRAINING MEMORY FOOTPRINT AND RUNTIME

Table 5 reports theoretical complexity in terms of standard matrix operations. We write ORTHO$(r, m)$ the complexity of orthogonalizing a family of $r$ vectors of dimension $m$. In our implemention, similar to powerSGD (Vogels et al., 2020), we apply power iteration the Gram-Schmidt process which has complexity $\mathcal{O}(r^2 * m)$.

Table 5: Optimizer comparison: matrix operation count for a weight layer of shape $n \times m$ with $n \leq m$ (i.e., left projection).

|  | Adam | LDAdam |
|---|---|---|
| Learning Subspace Adaptation |  | $n * r * m + 3 * n * m + r * m + \text{ORTHO}(r, m)$ |
| Projection-aware Update |  | $3 * r^2 * m + 4 * r * m + r^2$ |
| Adam-type Update | $9 * n * m$ | $9 * r * m$ |
| Descent Direction | $3 * n * m$ | $n * r * m + 3 * n * m$ |
| Error Buffer loading |  | $n * r * m + 3 * n * m$ |

Table 6 reports peak memory for fine-tuning and pre-training on a single NVIDIA H100 80GB GPU with micro batch size 1 and without activation checkpointing.

Table 6: Peak memory for fine-tuning (FT) and pre-training (PT) tasks for micro batch size of 1.

| Model | Task | Rank | Optimizer | |
|---|---|---|---|---|
|  |  |  | Adam | LDAdam |
| **RoBERTa-base** | **MNLI (FT)** | r=8 | 2.39 GB | 1.60 GB |
| **Llama 350M** | **C4 (PT)** | r=8 | 4.20 GB | 2.67 GB |
| **Llama 350M** | **C4 (PT)** | r=256 | 4.20 GB | 3.01 GB |
| **Llama 1.3B** | **C4 (PT)** | r=16 | 15.20 GB | 8.67 GB |
| **Llama-2 7B** | **GSM8K (FT)** | r=32 | 57.62 GB | 31.98 GB |
| **Llama-2 7B** | **GSM8K (FT)** | r=512 | 57.62 GB | 35.90 GB |

Figure 2 depicts the pre-training dynamics over time, allowing comparison of training efficiency with respect to time.

Figure 2: Pre-training dynamics over time for Llama 350M (left) and Llama 1.3B (right) on the C4 dataset.

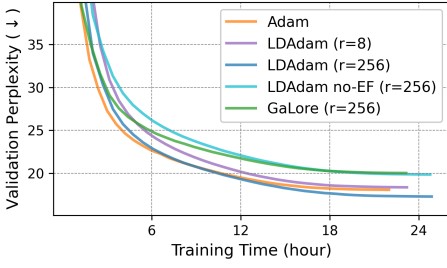 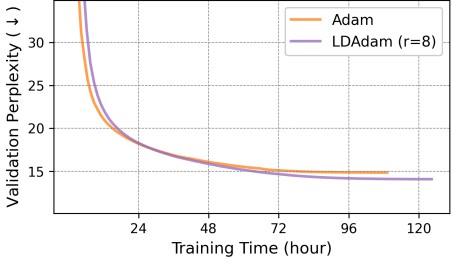

Figure 3 shows the throughput (token per second) and peak memory (GB) of Adam and LDAdam with respect to rank for pre-training the Llamma 350M model on the C4 dataset. We used a micro batch size of 1 for both to replicate the memory constrained setting. This helps to isolate the effect of the optimisation algorithm itself. See Table 7 and Table 6 for results closer to the standard use case.

Figure 3: Throughput (token per second) and peak memory (GB) of Adam and LDAdam with respect to rank for pre-training the Llamma 350M model on the C4 dataset, on a single NVIDIA H100 80BG GPU, using micro batch size of 1.

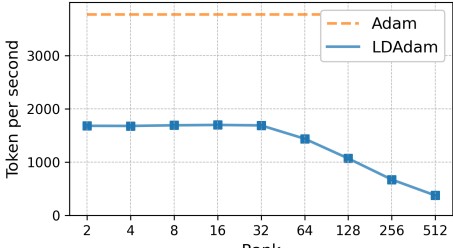 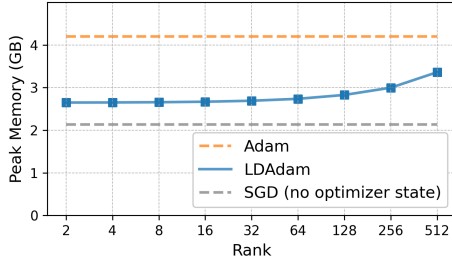

Table 7 reports runtime for fine-tuning and pre-training on a single (except for Llama 1.3B pre-training were we use four) NVIDIA H100 80GB GPU. We don't use activation checkpointing. We use a micro batch size of 1 for GSM8K fine-tuning, a micro batch size of 128 for Llama 350M model, and micro batch size of 64 for Llama 1.3B model.

Table 7: Runtime for fine-tuning (FT) and pre-training (PT) tasks.

| Model | Task | Rank | Optimizer | | | |
|---|---|---|---|---|---|---|
| | | | Adam | LDAdam | LDAdam no-EF | GaLore |
| **RoBERTa-base** | **MNLI (FT)** | r=8 | 37m | 01h 07m | 01h 01m | 56m |
| **Llama 350M** | **C4 (PT)** | r=8 | 21h 59m | 23h 12m | - | - |
| **Llama 350M** | **C4 (PT)** | r=256 | 21h 59m | 24h 50m | 24h 44m | 23h 09m |
| **Llama 1.3B** | **C4 (PT)** | r=16 | 04d 13h 09m | 05d 04h 24m | - | - |
| **Llama-2 7B** | **GSM8K (FT)** | r=32 | 50m | 56m | 55m | 01h 08m |
| **Llama-2 7B** | **GSM8K (FT)** | r=512 | 50m | 01h 08m | 01h 07m | 01h 09m |

## B.2 IMPACT OF THE RANK ON TRAINING

Figure 4: Training dynamics and validation perplexity for various rank when pre-training Llama 350M model. For training dynamics we used a single learning rate of $5e-4$ to allow comparison between runs and provide results for the first 10000 optimization steps. We report the best validation perplexity for learning rates tuned over the set $\{5e-4, 1e-3, 5e-3\}$.

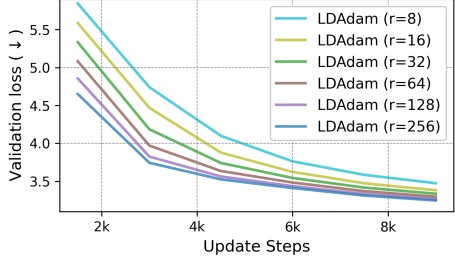 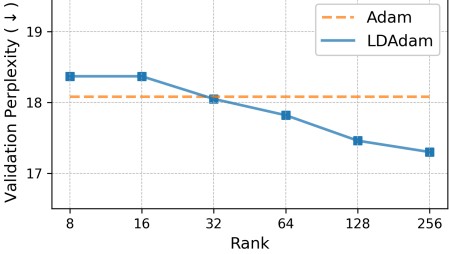

### B.3 FINE-TUNING STANDARD DEVIATION

Table 8: Standard deviation over 3 seeds for the results of fine-tuning RoBERTa-base model on the GLUE benchmark.

| Training Samples | MRPC 3.7k | STS-B 7k | CoLA 8.5k | SST-2 67k | QNLI 105k | QQP 364k | MNLI 393k |
|---|---|---|---|---|---|---|---|
| Adam | 1.068 | 0.226 | 2.604 | 0.517 | 0.074 | 0.050 | 0.167 |
| LDAdam | 0.617 | 0.211 | 0.633 | 0.463 | 0.275 | 0.080 | 0.177 |
| LDAdam no-EF | 0.849 | 0.555 | 0.903 | 0.066 | 0.285 | 0.687 | 0.087 |
| GaLore | 0.861 | 0.330 | 2.348 | 0.413 | 0.166 | 0.034 | 0.111 |

### B.4 ERROR BUFFER NORM DURING TRAINING

Figure 5: Error buffer norm and gradient norm during the fine-tuning of the RoBERTa-base model on the GLUE benchmark.

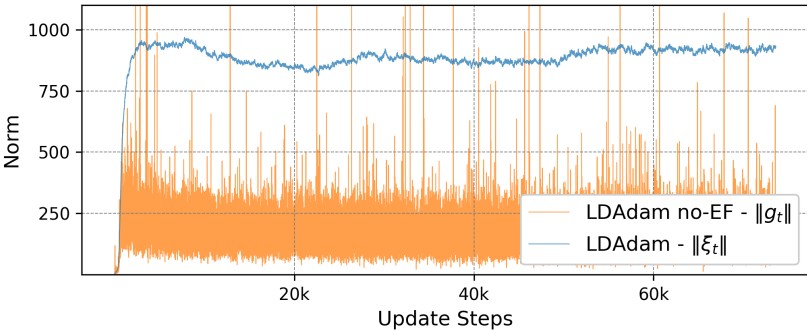

# C   MATERIALS FOR REPRODUCIBILITY

## C.1   OPTIMIZER STATES MEMORY ESTIMATES

Table 9 describes the architecture of the trained models architecture and the low-rank structure of the low-dimensional optimizer states. It allows optimizer states token count, thus memory estimates in GB by applying factor $\frac{2}{1024^3}$ for half precision training.

Table 9: Weights and low-dimensional optimizer states shape for used models.

|  | Weight | Low-rank optimizer states |
|---|---|---|
| **RoBERTa-base** | | |
| Token Embedding | 50265*768+768 | - |
| Positional Embedding | 564*768 | - |
| Attention Head | 12*4*768*768 | 12*4*3*768*r |
| MLP Block | 12*3*3072*768 | 12*3*(2*3072*r+768*r) |
| Normalization | 2*768+12*(9*768+3072) | - |
| Dense Layer | 768*768+768 | - |
| Output Layer | 768*n_label+n_label | - |
| **Llama 130M** | | |
| Embedding Layer | 32000*768 | - |
| Attention Head | 12*4*768*768 | 12*4*3*768*r |
| MLP Block | 12*3*2048*768 | 12*3*(2*2048*r+768*r) |
| Layer Normalization | (12*2+1)*768 | - |
| Output Layer | 768*32000 | - |
| **Llama 350M** | | |
| Embedding Layer | 32000*1024 | - |
| Attention Head | 24*4*1024*1024 | 24*4*3*1024*r - |
| MLP Block | 24*3*2736*1024 | 24*3*(2*2736*r+1024*r) |
| Layer Normalization | (24*2+1)*1024 | - |
| Output Layer | 1024*32000 | |
| **Llama-2 7B** | | |
| Embedding Layer | 32000*4096 | - |
| Attention Head | 32*4*4096*4096 | 32*4*3*4096*r |
| MLP Block | 32*3*4096*11008 | 32*3*(2*11008*r+4096*r) |
| Layer Normalization | (32*2+1)*4096 | - |
| Output Layer | 4096*32000 | - |

## C.2   FINE-TUNING HYPERPARAMETERS

Tables 10 and 11 detail all the hyperparameters we use when fine-tuning respectively RoBERTa-base model on the GLUE benchmark and Llama-2 7B model on the GSM8K dataset.

Table 10: Hyperparameters used for fine-tuning RoBERTa-base model on the GLUE benchmark.

| | Adam | LDAdam | GaLore |
|---|---|---|---|
| Epochs | | 3 | |
| Warm-up | | ✗ | |
| Batch Size | | 16 | |
| Maximum Length | | 128 | |
| Data Type | | `bfloat32` | |
| Learning Rate | | $\{1e-5, 2e-5, \ldots, 5e-5\}$ | |
| Learning Rate Scheduling | | linear to 0% | |
| Decay Rate $\beta_1$ | 0.9 | 0.908 | 0.9 |
| Decay Rate $\beta_2$ | 0.999 | 0.99 | 0.999 |
| Weight Decay | ✗ | ✗ | ✗ |
| Dropout | ✗ | ✗ | ✗ |
| Gradient Clipping | ✗ | ✗ | ✗ |
| Interpolation Factor $\rho$ | ✗ | 0.908 | ✗ |
| Error Feedback | ✗ | ✓ | ✗ |
| Subspace Frequency | ✗ | 1 | 500 |
| Learning Rate Scaling | ✗ | ✗ | ✗ |

Table 11: Hyperparameters used for fine-tuning Llama-2 7B model on the GSM8K dataset.

| | Adam | LDAdam | GaLore |
|---|---|---|---|
| Epochs | | 3 | |
| Training Steps | | 702 | |
| Warm-up Steps | | 20 | |
| Batch Size | | 32 | |
| Maximum Length | | 512 | |
| Data Type | | `bfloat16` | |
| Learning Rate | | $\{5e-5, 6e-5, \ldots, 9e-5\}$ | |
| | | $\{1e-4, 2e-4, \ldots, 5e-4\}$ | |
| Warm-up Scheduling | | linear from 0% | |
| Learning Rate Scheduling | | linear to 0% | |
| Decay Rate $\beta_1$ | 0.9 | 0.908 | 0.9 |
| Decay Rate $\beta_2$ | 0.999 | 0.99 | 0.999 |
| Weight Decay | ✗ | ✗ | ✗ |
| Dropout | ✗ | ✗ | ✗ |
| Gradient Clipping | 1.0 | ✗ | ✗ |
| Interpolation Factor $\rho$ | ✗ | 0.908 | ✗ |
| Error Feedback | ✗ | ✓ | ✗ |
| Subspace Frequency | ✗ | 1 | 200 |
| Learning Rate Scaling | ✗ | ✗ | ✗ |

### C.3  PRE-TRAINING HYPERPARAMETERS

Table 12 details all the hyperparameters we use when pre-training Llama models on the C4 dataset.

Table 12: Hyperparameters used for pre-training Llama models on the C4 dataset.

| | Llama 130M | | | Llama 350M | | | Llama 1.3B | |
| --- | --- | --- | --- | --- | --- | --- | --- | --- |
| | Adam | LDAdam | GaLore | Adam | LDAdam | GaLore | Adam | LDAdam |
| Training Steps | 20000 | | | 55000 | | | 200000 | |
| Warm-up Steps | 2000 | | | 5500 | | | 10000 | |
| Maximum Length | 256 | | | 256 | | | 256 | |
| Batch Size | 512 | | | 512 | | | 512 | |
| Token Batch Size | 131 072 | | | 131 072 | | | 131 072 | |
| Total Training Tokens | 2 621 440 000 | | | 7 208 960 000 | | | 26 621 440 000 | |
| Data Type | bfloat16 | | | bfloat16 | | | bfloat16 | |
| Learning Rate | $\{5e-4, 1e-3, 5e-3\}$ | | | $\{5e-4, 1e-3, 5e-3\}$ | | | $5e-4$ | |
| Warm-up Scheduling | linear from 0% | | | linear from 0% | | | linear from 0% | |
| Learning Rate Scheduling | cosine to 10% | | | cosine to 10% | | | cosine to 10% | |
| Decay Rate $\beta_1$ | 0.9 | 0.908 | 0.9 | 0.9 | 0.908 | 0.9 | 0.9 | 0.908 |
| Decay Rate $\beta_2$ | 0.999 | 0.98 | 0.999 | 0.999 | 0.98 | 0.999 | 0.999 | 0.98 |
| Weight Decay | ✗ | ✗ | ✗ | ✗ | ✗ | ✗ | ✗ | ✗ |
| Dropout | ✗ | ✗ | ✗ | ✗ | ✗ | ✗ | ✗ | ✗ |
| Gradient Clipping | 1.0 | ✗ | ✗ | 1.0 | ✗ | ✗ | 1.0 | ✗ |
| Interpolation Factor $\rho$ | ✗ | 0.908 | ✗ | ✗ | 0.908 | ✗ | ✗ | 0.908 |
| Error Feedback | ✗ | ✓ | ✗ | ✗ | ✓ | ✗ | ✗ | ✓ |
| Subspace Frequency | ✗ | 1 | 200 | ✗ | 1 | 200 | ✗ | 1 |
| Learning Rate Scaling | ✗ | ✗ | ✗ | ✗ | ✗ | ✗ | ✗ | ✗ |

# D  GaLore Algorithm

Algorithm 2 describes GaLore's (Zhao et al., 2024) proposal for performing an Adam-type update from a low-rank gradient projection. For experiments we use the author's implementation of the GaLore algorithm accessible at: `https://github.com/jiaweizzhao/GaLore`. Other than layer-wise learning rate rescaling, in our experiments we use the suggested hyperparameters.

---

**Algorithm 2** GaLore

**Hyperparameters:** step size $\eta_t$; decay rates $\beta_1$, $\beta_2$
**GaLore Hyperparameters:** projection rank $r$; subspace change frequency $\mathcal{T}$, scale factor $\alpha$
1: **Initialization:** $m_0 = 0$; $v_0 = 0$; $\mathcal{U}_0 = 0$
2: **for** $t = \{1, 2, \ldots, T\}$ **do**
3:     **if** $t \mod \mathcal{T} = 0$ **then**
4:         $\bar{\mathcal{U}}_t, \Sigma, V = \text{SVD}(g_t)$
5:         $\mathcal{U}_t = \bar{\mathcal{U}}_t[:, 1:r]$
6:     **else**
7:         $\mathcal{U}_t = \mathcal{U}_{t-1}$
8:     **end if**
9:     $a_t = \mathcal{U}_t^\top g_t$
10:     $m_t = \beta_1 m_t + (1 - \beta_1)a_t$
11:     $v_t = \beta_2 v_t + (1 - \beta_2)a_t^2$
12:     $\theta_{t+1} = \theta_t - \alpha\eta_t\mathcal{U}_t \cdot \frac{\hat{m}_t}{\sqrt{\hat{v}_t}+\epsilon}$
13: **end for**

---

## D.1  layer-wise Learning Rate Rescaling

Note that the learning rate rescaling induced by multiplying by $\alpha \neq 1$ is equivalent to using different learning rates for different layers. This level of hyperparameter tuning is not usual, so for a fair comparison we use a single learning rate for all layers. However, our preliminary experiments suggest that LDAdam would also benefit from layer-wise learning rate tuning.

## D.2  Incompatibility of Gradient Accumulation and Per-layer Weight Update

GaLore's improvement in memory efficiency over the standard Adam implementation comes from three distinct additions, namely: low-rank gradient projection (Zhao et al., 2024), 8-bit quantization (Dettmers et al., 2022), and per-layer weight update (Lv et al., 2024b). The latter saves memory by releasing the variable used to store the gradient after the model layer has been updated rather than after the entire model has been updated. A practical implementation of per-layer weight updates is to add a gradient hook that triggers the model update and releases the gradient variable immediately after the gradient for that particular layer has been computed. For example, in PyTorch, the per-layer weight update is built by adding a gradient hook using `p.register_post_accumulate_grad_hook(optimizer_hook)`, with the `optimizer_hook` function implementing both `optimizer.step()` and `optimizer.zero_grad()` (Pytorch_Tutorials, n.d.).

Therefore, when using per-layer weight update, one never has access to the gradient for the entire model, and furthermore one cannot accumulate the gradient over multiple micro batches and perform the model update (e.g. in PyTorch, run `optimizer.step()`) only after accumulation. For the same reason, gradient clipping is not possible when using per-layer weight update, since the norm of the gradient for the entire model is not computable.

To enable gradient accumulation for GaLore, the per-layer weight update has to be abandoned. The method is called by the author GaLore (no retaining grad) (Zhao et al., 2024) and it results in an additional memory overhead equal to the size of the model compared to GaLore's claim.

# E   DEFERRED PROOFS

---
**Algorithm 3** LDAdam: Analytical View

---
1: **Hyperparameters:** step size $\eta_t$, decay rates $\beta_1$ and $\beta_2$, projection rank $r$ with contraction $q_r$.
2: **Initialization:** error $\xi_1 = 0_d$, moments $m_0 = v_0 = \tilde{v}_0 = 0_r$ and $\mathcal{U}_0 = 0_{d \times r}$.
3: **for** $t = \{1, 2, \ldots, T\}$ **do**
   **Compute error corrected gradient and momentum**
4:    $A_t = g_t + \xi_t$, where $g_t = \widetilde{\nabla}_\theta f(\theta_t)$ is a mini-batch stochastic gradient at $\theta_t$
5:    $B_t = \beta_1 \mathcal{U}_{t-1} m_{t-1} + (1 - \beta_1) A_t$
   **Update the projection matrix**
6:    $\mathcal{U}_t$ is any $d \times r$ orthogonal matrix such that $\|(I - \mathcal{U}_t \mathcal{U}_t^\top) B_t\| \le q_r \|B_t\|$ with $q_r < 1$
   **Intermediate updates to adjust to the new low-dimensional space**
7:    $a_t = \mathcal{U}_t^\top A_t$
8:    $m_{t-1/2} = \mathcal{U}_t^\top \mathcal{U}_{t-1} m_{t-1}$
9:    $v_{t-1/2} = (1 - \beta_2^{t-1}) \left| (\mathcal{U}_t^\top \mathcal{U}_{t-1})^2 \cdot \left( \frac{v_{t-1}}{1 - \beta_2^{t-1}} - (\frac{m_{t-1}}{1 - \beta_1^{t-1}})^2 \right) + (\mathcal{U}_t^\top \mathcal{U}_{t-1} \cdot \frac{m_{t-1}}{1 - \beta_1^{t-1}})^2 \right|$
   **Adam updates in the low-dimensional space**
10:   $m_t = \beta_1 m_{t-1/2} + (1 - \beta_1) a_t$
11:   $v_t = \beta_2 v_{t-1/2} + (1 - \beta_2) a_t^2$
12:   $\tilde{v}_t = \max(v_t, \|\tilde{v}_{t-1}\|_{\max})$                    $\diamond$ AMSGrad-type normalization
   **Update the main model**
13:   $\theta_{t+1} = \theta_t - \eta_t \mathcal{U}_t \cdot \frac{m_t}{\sqrt{\tilde{v}_t} + \epsilon}$
   **Update the error feedback**
14:   $\xi_{t+1} = (A_t - \mathcal{U}_t \cdot a_t) + \frac{\beta_1}{1 - \beta_1} (\mathcal{U}_{t-1} \cdot m_{t-1} - \mathcal{U}_t \cdot m_{t-1/2})$
15: **end for**

---

## E.1   KEY LEMMAS

**Lemma 1.** *With $\Sigma_T = T\sigma^2 + \sum_{t=1}^T \mathbb{E}\left[\|\nabla f(\theta_t)\|^2\right]$, for any $t \ge 1$ the following bounds hold:*

$$\|B_t\| \le \frac{G}{1 - q_r}, \quad \sum_{t=1}^T \mathbb{E}\left[\|B_t\|^2\right] \le \frac{1}{(1 - q_r)^2} \Sigma_T \tag{13}$$

$$\|\xi_t\| \le \frac{q_r G}{(1 - \beta_1)(1 - q_r)}, \quad \sum_{t=1}^T \mathbb{E}\left[\|\xi_t\|^2\right] \le \frac{q_r^2}{(1 - \beta_1)^2 (1 - q_r)^2} \Sigma_T \tag{14}$$

$$\left\| \frac{\beta_1}{1 - \beta_1} B_t + (1 - \beta_1) \xi_{t+1} \right\| \le \frac{\beta_1 + (1 - \beta_1) q_r}{(1 - \beta_1)(1 - q_r)} G \tag{15}$$

$$\sum_{t=1}^T \mathbb{E}\left[ \left\| \frac{\beta_1}{1 - \beta_1} B_{t-1} + (1 - \beta_1) \xi_t \right\|^2 \right] \le \frac{\beta_1 + (1 - \beta_1) q_r^2}{(1 - \beta_1)^2 (1 - q_r)^2} \Sigma_T. \tag{16}$$

*Proof.* Let us start with the proof of the first bound on $B_t$. Denote $\tilde{q}_r := (1 - \beta_1) q_r + \beta_1 \in [q_r, 1)$.

$$
\begin{aligned}
\|B_{t+1}\| &= \|\beta_1 M_t + (1 - \beta_1) A_{t+1}\| \\
&= \|\beta_1 M_t + (1 - \beta_1) g_{t+1} + (1 - \beta_1) \xi_{t+1}\| \\
&= \|-(1 - \beta_1) M_t + (1 - \beta_1) g_{t+1} + B_t\| \\
&= \|-(1 - \beta_1) \mathcal{U}_t \mathcal{U}_t^\top B_t + (1 - \beta_1) g_{t+1} + B_t\| \\
&= \|(1 - \beta_1)(B_t - \mathcal{U}_t \mathcal{U}_t^\top B_t) + (1 - \beta_1) g_{t+1} + \beta_1 B_t\| \\
&\le (1 - \beta_1) \|B_t - \mathcal{U}_t \mathcal{U}_t^\top B_t\| + (1 - \beta_1) \|g_{t+1}\| + \beta_1 \|B_t\| \\
&\le ((1 - \beta_1) q_r + \beta_1) \|B_t\| + (1 - \beta_1) \|g_{t+1}\| \\
&= \tilde{q}_r \|B_t\| + (1 - \beta_1) \|g_{t+1}\| \\
&= \tilde{q}_r^t \|B_1\| + (1 - \beta_1) \sum_{\tau=2}^{t+1} \tilde{q}_r^{t+1-\tau} \|g_\tau\| = (1 - \beta_1) \sum_{\tau=1}^{t+1} \tilde{q}_r^{t+1-\tau} \|g_\tau\|.
\end{aligned}
$$

Using the bounded gradient assumption, we get

$$\|B_t\| \leq (1-\beta_1)G \sum_{\tau=1}^{t} \tilde{q}_r^{t-\tau} \leq \frac{(1-\beta_1)G}{1-\tilde{q}_r} = \frac{G}{1-q_r}.$$

To derive the bound with expectation, we apply Cauchy-Schwartz inequality and the bounded variance assumption:

$$
\begin{aligned}
\sum_{t=1}^{T} \mathbb{E}\left[\|B_t\|^2\right] &\leq (1-\beta_1)^2 \sum_{t=1}^{T} \mathbb{E}\left[\left(\sum_{\tau=1}^{t} \tilde{q}_r^{t-\tau}\|g_\tau\|\right)^2\right] \\
&\leq (1-\beta_1)^2 \sum_{t=1}^{T} \mathbb{E}\left[\left(\sum_{\tau=1}^{t} \tilde{q}_r^{t-\tau}\right)\left(\sum_{\tau=1}^{t} \tilde{q}_r^{t-\tau}\|g_\tau\|^2\right)\right] \\
&\leq \frac{(1-\beta_1)^2}{1-\tilde{q}_r} \sum_{t=1}^{T} \sum_{\tau=1}^{t} \tilde{q}_r^{t-\tau} \mathbb{E}\left[\|g_\tau\|^2\right] \\
&\leq \frac{(1-\beta_1)^2}{(1-\tilde{q}_r)^2} \sum_{t=1}^{T} \mathbb{E}\left[\|g_t\|^2\right] \\
&= \frac{1}{(1-q_r)^2} \sum_{t=1}^{T} \mathbb{E}\left[\|g_t - \nabla f(\theta_t) + \nabla f(\theta_t)\|^2\right] \\
&\leq \frac{1}{(1-q_r)^2} \sum_{t=1}^{T} \left(\sigma^2 + \mathbb{E}\left[\|\nabla f(\theta_t)\|^2\right]\right) \\
&\leq \frac{T\sigma^2}{(1-q_r)^2} + \frac{1}{(1-q_r)^2} \sum_{t=1}^{T} \mathbb{E}\left[\|\nabla f(\theta_t)\|^2\right]
\end{aligned}
$$

To bound the norm of the error $\|\xi_t\|$, notice that

$$\|\xi_{t+1}\| = \frac{1}{1-\beta_1}\|B_t - \mathcal{U}_t\mathcal{U}_t^\top B_t\| \leq \frac{q_r}{1-\beta_1}\|B_t\| \leq \frac{q_r G}{(1-\beta_1)(1-q_r)}. \tag{17}$$

To get the bound with expectation, we apply previous inequality on $B_t$ and get

$$
\begin{aligned}
\sum_{t=1}^{T} \mathbb{E}\left[\|\xi_t\|^2\right] &= \sum_{t=1}^{T-1} \mathbb{E}\left[\|\xi_{t+1}\|^2\right] \\
&\leq \sum_{t=1}^{T-1} \frac{q_r^2}{(1-\beta_1)^2} \mathbb{E}\left[\|B_t\|^2\right] \\
&\leq \frac{q_r^2}{(1-\beta_1)^2(1-q_r)^2}\left(T\sigma^2 + \sum_{t=1}^{T} \mathbb{E}\left[\|\nabla f(\theta_t)\|^2\right]\right)
\end{aligned}
$$

The fifth bound (15) follows from the triangle inequality and combining the obtained two bounds. For the last bound with expectation, we have

$$
\begin{aligned}
\sum_{t=1}^{T} \mathbb{E}\left[\left\|\frac{\beta_1}{1-\beta_1}B_{t-1} + (1-\beta_1)\xi_t\right\|^2\right] &\leq \sum_{t=1}^{T}\left(\frac{\beta_1}{(1-\beta_1)^2}\mathbb{E}\left[\|B_{t-1}\|^2\right] + (1-\beta_1)\mathbb{E}\left[\|\xi_t\|^2\right]\right) \\
&\leq \frac{\beta_1}{(1-\beta_1)^2}\left(\frac{T\sigma^2}{(1-q_r)^2} + \frac{1}{(1-q_r)^2}\sum_{t=1}^{T}\mathbb{E}\left[\|\nabla f(\theta_t)\|^2\right]\right) \\
&\quad + \frac{q_r^2}{(1-\beta_1)(1-q_r)^2}\left(T\sigma^2 + \sum_{t=1}^{T}\mathbb{E}\left[\|\nabla f(\theta_t)\|^2\right]\right) \\
&= \frac{\beta_1 + (1-\beta_1)q_r^2}{(1-\beta_1)^2(1-q_r)^2}\left(T\sigma^2 + \sum_{t=1}^{T}\mathbb{E}\left[\|\nabla f(\theta_t)\|^2\right]\right).
\end{aligned}
$$

$\square$

**Lemma 2.** *If $\gamma < 1$ and $1 - \gamma \le \frac{1}{2}(1 - \beta_1)(1 - q_r)$, then*

$$\sum_{t=1}^{T} \gamma^{T-t} \mathbb{E}\left[\left\|\frac{\beta_1}{1 - \beta_1} B_{t-1} + (1 - \beta_1)\xi_t\right\|^2\right] \le 2C_2 \left(\frac{\sigma^2}{1 - \gamma} + \sum_{t=1}^{T} \gamma^{T-t} \mathbb{E}\left[\|\nabla f(\theta_t)\|^2\right]\right), \quad (18)$$

*where $C_2 = \frac{\beta_1 + (1 - \beta_1)q_r^2}{(1 - \beta_1)^2(1 - q_r)^2}$.*

*Proof.* From the condition on $\gamma$ and the notation $\tilde{q}_r := (1 - \beta_1)q_r + \beta_1 \in [q_r, 1)$ from the previous proof, we have $1 - \gamma \le \frac{1 - \tilde{q}_r}{2}$ or equivalently $\gamma - \tilde{q}_r \ge \frac{1 - \tilde{q}_r}{2}$. Using this inequality on $\gamma$ and the previous bound (16) for $\mathbb{E}\left[\|B_t\|^2\right]$, we have

$$
\begin{aligned}
\sum_{t=1}^{T} \gamma^{T-t} \mathbb{E}\left[\|B_t\|^2\right] &\le (1 - \beta_1)^2 \sum_{t=1}^{T} \gamma^{T-t} \mathbb{E}\left[\left(\sum_{\tau=1}^{t} \tilde{q}_r^{t-\tau} \|g_\tau\|\right)^2\right] \\
&\le (1 - \beta_1)^2 \sum_{t=1}^{T} \gamma^{T-t} \mathbb{E}\left[\left(\sum_{\tau=1}^{t} \tilde{q}_r^{t-\tau}\right)\left(\sum_{\tau=1}^{t} \tilde{q}_r^{t-\tau} \|g_\tau\|^2\right)\right] \\
&\le \frac{(1 - \beta_1)^2}{1 - \tilde{q}_r} \sum_{t=1}^{T} \sum_{\tau=1}^{t} \gamma^{T-t} \tilde{q}_r^{t-\tau} \mathbb{E}\left[\|g_\tau\|^2\right] \\
&= \frac{(1 - \beta_1)^2}{1 - \tilde{q}_r} \sum_{\tau=1}^{T} \sum_{t=\tau}^{T} \gamma^{T-t} \tilde{q}_r^{t-\tau} \mathbb{E}\left[\|g_\tau\|^2\right] \\
&= \frac{(1 - \beta_1)^2}{1 - \tilde{q}_r} \sum_{\tau=1}^{T} \left(\sum_{t=\tau}^{T} \gamma^{\tau-t} \tilde{q}_r^{t-\tau}\right) \gamma^{T-\tau} \mathbb{E}\left[\|g_\tau\|^2\right] \\
&= \frac{(1 - \beta_1)^2}{1 - \tilde{q}_r} \sum_{\tau=1}^{T} \frac{1 - (\tilde{q}_r/\gamma)^{T-\tau+1}}{1 - \tilde{q}_r/\gamma} \gamma^{T-\tau} \mathbb{E}\left[\|g_\tau\|^2\right] \\
&\le \frac{(1 - \beta_1)^2}{(1 - \tilde{q}_r)(1 - \tilde{q}_r/\gamma)} \sum_{t=1}^{T} \gamma^{T-\tau} \mathbb{E}\left[\|g_t\|^2\right] \\
&\le \frac{2(1 - \beta_1)^2}{(1 - \tilde{q}_r)^2} \sum_{t=1}^{T} \gamma^{T-\tau} \mathbb{E}\left[\|g_t\|^2\right] \\
&= \frac{2}{(1 - q_r)^2} \sum_{t=1}^{T} \gamma^{T-\tau} \mathbb{E}\left[\|g_t - \nabla f(\theta_t) + \nabla f(\theta_t)\|^2\right] \\
&\le \frac{2}{(1 - q_r)^2} \sum_{t=1}^{T} \gamma^{T-\tau} \left(\sigma^2 + \mathbb{E}\left[\|\nabla f(\theta_t)\|^2\right]\right) \\
&\le \frac{2}{(1 - q_r)^2} \left(\frac{\sigma^2}{1 - \gamma} + \sum_{t=1}^{T} \gamma^{T-\tau} \mathbb{E}\left[\|\nabla f(\theta_t)\|^2\right]\right).
\end{aligned}
$$

Using the bound (17) for $\|\xi_t\|^2$, we have

$$\sum_{t=1}^{T} \gamma^{T-\tau} \mathbb{E}\left[\|\xi_t\|^2\right] = \frac{2q_r^2}{(1 - \beta_1)^2(1 - q_r)^2} \left(\frac{\sigma^2}{1 - \gamma} + \sum_{t=1}^{T} \gamma^{T-\tau} \mathbb{E}\left[\|\nabla f(\theta_t)\|^2\right]\right).$$

Combining these two bounds we get

$$\sum_{t=1}^{T} \gamma^{T-t} \mathbb{E}\left[\left\|\frac{\beta_1}{1-\beta_1}B_{t-1} + (1-\beta_1)\xi_t\right\|^2\right]$$

$$\leq \sum_{t=1}^{T} \gamma^{T-t}\left(\frac{\beta_1}{(1-\beta_1)^2}\mathbb{E}\left[\|B_{t-1}\|^2\right] + (1-\beta_1)\mathbb{E}\left[\|\xi_t\|^2\right]\right)$$

$$\leq \frac{\beta_1}{(1-\beta_1)^2}\frac{2}{(1-q_r)^2}\left(\frac{\sigma^2}{1-\gamma} + \sum_{t=1}^{T}\gamma^{T-t}\mathbb{E}\left[\|\nabla f(\theta_t)\|^2\right]\right)$$

$$+\frac{2(1-\beta_1)q_r^2}{(1-\beta_1)^2(1-q_r)^2}\left(\frac{\sigma^2}{1-\gamma} + \sum_{t=1}^{T}\gamma^{T-t}\mathbb{E}\left[\|\nabla f(\theta_t)\|^2\right]\right)$$

$$=\frac{2(\beta_1 + (1-\beta_1)q_r^2)}{(1-\beta_1)^2(1-q_r)^2}\left(\frac{\sigma^2}{1-\gamma} + \sum_{t=1}^{T}\gamma^{T-t}\mathbb{E}\left[\|\nabla f(\theta_t)\|^2\right]\right).$$

$\square$

**Lemma 3.** *For $\Delta\Gamma_t = \Gamma_{t-1} - \Gamma_t$ we have*

$$\sum_{t=1}^{T}\|\Delta\Gamma_t\| \leq \frac{2}{\sqrt{\epsilon}}, \quad \sum_{t=1}^{T}\|\Delta\Gamma_t\|^2 \leq \frac{2}{\epsilon}.$$

*Proof.* From the definitions of $\Gamma_t$ (19) and $\tilde{v}_t = \max(v_t, \|\tilde{v}_{t-1}\|_{\max})$ we have

$$\begin{aligned}
\Gamma_t &= \bar{\mathcal{U}}_t \mathrm{Diag}^{-1/2}(\tilde{v}_t + \epsilon, \|\tilde{v}_t\|_{\min} + \epsilon)\bar{\mathcal{U}}_t^\top \\
&\preceq \bar{\mathcal{U}}_t \mathrm{Diag}^{-1/2}(\|\tilde{v}_t\|_{\min} + \epsilon)\bar{\mathcal{U}}_t^\top \\
&= \frac{1}{\sqrt{\|\tilde{v}_t\|_{\min} + \epsilon}}\bar{\mathcal{U}}_t\bar{\mathcal{U}}_t^\top = \frac{1}{\sqrt{\|\tilde{v}_t\|_{\min} + \epsilon}}I \\
&\preceq \frac{1}{\sqrt{\|\tilde{v}_{t-1}\|_{\max} + \epsilon}}I = \frac{1}{\sqrt{\|\tilde{v}_{t-1}\|_{\max} + \epsilon}}\bar{\mathcal{U}}_{t-1}\bar{\mathcal{U}}_{t-1}^\top \\
&= \bar{\mathcal{U}}_{t-1}\mathrm{Diag}^{-1/2}(\|\tilde{v}_{t-1}\|_{\max} + \epsilon)\bar{\mathcal{U}}_{t-1}^\top \\
&\preceq \bar{\mathcal{U}}_{t-1}\mathrm{Diag}^{-1/2}(\tilde{v}_{t-1} + \epsilon, \|\tilde{v}_{t-1}\|_{\min} + \epsilon)\bar{\mathcal{U}}_{t-1}^\top = \Gamma_{t-1},
\end{aligned}$$

which implies that $\Delta\Gamma_t = \Gamma_{t-1} - \Gamma_t$ is positive semidefinite. Hence, $\|\Delta\Gamma_t\| = \lambda_{\max}(\Delta\Gamma_t) \geq 0$. Using the convexity of $\lambda_{\max}$ over symmetric matrices, we get

$$\begin{aligned}
\sum_{t=1}^{T}\|\Delta\Gamma_t\| &= \sum_{t=1}^{T}\lambda_{\max}(\Gamma_{t-1} - \Gamma_t) \\
&\leq \sum_{t=1}^{T}\lambda_{\max}(\Gamma_{t-1}) + \lambda_{\max}(-\Gamma_t) = \sum_{t=1}^{T}\lambda_{\max}(\Gamma_{t-1}) - \lambda_{\min}(\Gamma_t) \\
&= \sum_{t=1}^{T}\frac{1}{\sqrt{\|\tilde{v}_{t-1}\|_{\min} + \epsilon}} - \frac{1}{\sqrt{\|\tilde{v}_t\|_{\max} + \epsilon}} \\
&= \sum_{t=1}^{T}\frac{1}{\sqrt{\|\tilde{v}_{t-1}\|_{\min} + \epsilon}} - \frac{1}{\sqrt{\|\tilde{v}_{t-1}\|_{\max} + \epsilon}} + \frac{1}{\sqrt{\|\tilde{v}_{t-1}\|_{\max} + \epsilon}} - \frac{1}{\sqrt{\|\tilde{v}_t\|_{\max} + \epsilon}} \\
&\leq \sum_{t=1}^{T}\frac{1}{\sqrt{\|\tilde{v}_{t-1}\|_{\min} + \epsilon}} - \frac{1}{\sqrt{\|\tilde{v}_t\|_{\min} + \epsilon}} + \sum_{t=1}^{T}\frac{1}{\sqrt{\|\tilde{v}_{t-1}\|_{\max} + \epsilon}} - \frac{1}{\sqrt{\|\tilde{v}_t\|_{\max} + \epsilon}} \\
&= \frac{1}{\sqrt{\|\tilde{v}_0\|_{\min} + \epsilon}} - \frac{1}{\sqrt{\|\tilde{v}_T\|_{\min} + \epsilon}} + \frac{1}{\sqrt{\|\tilde{v}_0\|_{\max} + \epsilon}} - \frac{1}{\sqrt{\|\tilde{v}_T\|_{\max} + \epsilon}} \\
&\leq \frac{2}{\sqrt{\|\tilde{v}_0\|_{\min} + \epsilon}} \leq \frac{2}{\sqrt{\epsilon}}.
\end{aligned}$$

For the second sum of squared norms, notice that for scalars $a \geq b \geq 0$, it holds that

$$(a - b)^2 \leq (a - b)(a + b) = a^2 - b^2.$$

Therefore, the above derivation can be repeated without the square roots as follows:

$$
\begin{aligned}
\sum_{t=1}^{T} \|\Delta\Gamma_t\|^2 &= \sum_{t=1}^{T} (\lambda_{\max}(\Gamma_{t-1} - \Gamma_t))^2 \\
&\leq \sum_{t=1}^{T} (\lambda_{\max}(\Gamma_{t-1}) + \lambda_{\max}(-\Gamma_t))^2 = \sum_{t=1}^{T} (\lambda_{\max}(\Gamma_{t-1}) - \lambda_{\min}(\Gamma_t))^2 \\
&\leq \sum_{t=1}^{T} (\lambda_{\max}(\Gamma_{t-1}))^2 - (\lambda_{\min}(\Gamma_t))^2 \\
&= \sum_{t=1}^{T} \frac{1}{\|\tilde{v}_{t-1}\|_{\min} + \epsilon} - \frac{1}{\|\tilde{v}_t\|_{\max} + \epsilon} \\
&= \sum_{t=1}^{T} \frac{1}{\|\tilde{v}_{t-1}\|_{\min} + \epsilon} - \frac{1}{\|\tilde{v}_{t-1}\|_{\max} + \epsilon} + \frac{1}{\|\tilde{v}_{t-1}\|_{\max} + \epsilon} - \frac{1}{\|\tilde{v}_t\|_{\max} + \epsilon} \\
&\leq \sum_{t=1}^{T} \frac{1}{\|\tilde{v}_{t-1}\|_{\min} + \epsilon} - \frac{1}{\|\tilde{v}_t\|_{\min} + \epsilon} + \sum_{t=1}^{T} \frac{1}{\|\tilde{v}_{t-1}\|_{\max} + \epsilon} - \frac{1}{\|\tilde{v}_t\|_{\max} + \epsilon} \\
&= \frac{1}{\|\tilde{v}_0\|_{\min} + \epsilon} - \frac{1}{\|\tilde{v}_T\|_{\min} + \epsilon} + \frac{1}{\|\tilde{v}_0\|_{\max} + \epsilon} - \frac{1}{\|\tilde{v}_T\|_{\max} + \epsilon} \\
&\leq \frac{2}{\|\tilde{v}_0\|_{\min} + \epsilon} \leq \frac{2}{\epsilon},
\end{aligned}
$$

which completes the proof. $\qquad\square$

**Lemma 4.** *For all iterates $t \geq 1$ the following bound holds*

$$\|\tilde{v}_t\|_{\max} \leq \frac{1 + \beta_2}{1 - \beta_2} \frac{(1 - \beta_1(1 - q_r))^2}{(1 - \beta_1)^2 (1 - q_r)^2} G^2.$$

*Proof.* First, let us bound the $a_t$ term using Lemma 1 and Assumption 2.

$$\|a_t\| = \|\mathcal{U}_t^\top A_t\| \leq \|A_t\| \leq \|\xi_t\| + \|g_t\| \leq \frac{1 - (1 - q_r)\beta_1}{(1 - \beta_1)(1 - q_r)} G =: CG.$$

Next, we bound momentum $m_t$:

$$
\begin{aligned}
\|m_t\| &\leq \beta_1 \|m_{t-1/2}\| + (1 - \beta_1)\|a_t\| \\
&\leq \beta_1 \|m_{t-1}\| + (1 - \beta_1) CG \\
&\leq \beta_1^t \|m_0\| + (1 - \beta_1) CG \sum_{\tau=0}^{t-1} \beta_1^\tau = (1 - \beta_1^t) CG.
\end{aligned}
$$

Next, we bound the intermediate term $\|v_{t-1/2}\|_1$. Note that by the triangle inequality we have the following direct bound for it:

$$\left\| (\mathcal{U}_t^\top \mathcal{U}_{t-1})^2 v_{t-1} \right\|_1 + \left\| (\mathcal{U}_t^\top \mathcal{U}_{t-1})^2 \left( \frac{m_{t-1}}{1 - \beta_1^{t-1}} \right)^2 \right\|_1 + \left\| \left( \mathcal{U}_t^\top \mathcal{U}_{t-1} \cdot \frac{m_{t-1}}{1 - \beta_1^{t-1}} \right)^2 \right\|_1$$

Now let us bound each term separately. For the first term we have

$$
\begin{aligned}
\left\|\left(\mathcal{U}_t^\top \mathcal{U}_{t-1}\right)^2 v_{t-1}\right\|_1 &= \sum_{i=1}^r \sum_{j=1}^r \langle \mathcal{U}_t[:,i], \mathcal{U}_{t-1}[:,j]\rangle^2 v_{t-1,j} \\
&\leq \sum_{j=1}^r v_{t-1,j} \cdot \max_{1\leq j\leq r} \sum_{i=1}^r \langle \mathcal{U}_t[:,i], \mathcal{U}_{t-1}[:,j]\rangle^2 \\
&= \|v_{t-1}\|_1 \max_{1\leq j\leq r} \left\|\mathcal{U}_t^\top \mathcal{U}_{t-1}[:,j]\right\|^2 \\
&\leq \|v_{t-1}\|_1 \max_{1\leq j\leq r} \|\mathcal{U}_{t-1}[:,j]\|^2 = \|v_{t-1}\|_1,
\end{aligned}
$$

since columns $\mathcal{U}_t[:,i]$ and $\mathcal{U}_{t-1}[:,j]$ have unit length by construction. Similarly, for the second term we have

$$
\begin{aligned}
\left\|\left(\mathcal{U}_t^\top \mathcal{U}_{t-1}\right)^2 \left(\frac{m_{t-1}}{1-\beta_1^{t-1}}\right)^2\right\|_1 &= \sum_{i=1}^r \sum_{j=1}^r \langle \mathcal{U}_t[:,i], \mathcal{U}_{t-1}[:,j]\rangle^2 \left(\frac{m_{t-1,j}}{1-\beta_1^{t-1}}\right)^2 \\
&\leq \sum_{j=1}^r \left(\frac{m_{t-1,j}}{1-\beta_1^{t-1}}\right)^2 \cdot \max_{1\leq j\leq r} \sum_{i=1}^r \langle \mathcal{U}_t[:,i], \mathcal{U}_{t-1}[:,j]\rangle^2 \\
&= \left\|\frac{m_{t-1}}{1-\beta_1^{t-1}}\right\|^2 \max_{1\leq j\leq r} \left\|\mathcal{U}_t^\top \mathcal{U}_{t-1}[:,j]\right\|^2 \leq \left\|\frac{m_{t-1}}{1-\beta_1^{t-1}}\right\|^2 \leq C^2 G^2
\end{aligned}
$$

Finally, for the third term we have

$$
\left\|\left(\mathcal{U}_t^\top \mathcal{U}_{t-1} \cdot \frac{m_{t-1}}{1-\beta_1^{t-1}}\right)^2\right\|_1 = \left\|\mathcal{U}_t^\top \mathcal{U}_{t-1} \cdot \frac{m_{t-1}}{1-\beta_1^{t-1}}\right\|^2 \leq \left\|\frac{m_{t-1}}{1-\beta_1^{t-1}}\right\|^2 \leq C^2 G^2.
$$

Combining all three bounds together, we arrive

$$
\|v_{t-1/2}\|_1 \leq \|v_{t-1}\|_1 + 2C^2 G^2.
$$

From this we get the bound for $v_t$ using the initialization $v_0 = 0$:

$$
\begin{aligned}
\|v_t\|_{\max} \leq \|v_t\|_1 &\leq \beta_2 \|v_{t-1/2}\|_1 + (1-\beta_2)\|a_t\|^2 \\
&\leq \beta_2(\|v_{t-1}\|_1 + 2C^2 G^2) + (1-\beta_2)C^2 G^2 \\
&\leq \beta_2 \|v_{t-1}\|_1 + (1+\beta_2)C^2 G^2 \\
&\leq \beta_2^t \|v_0\|_1 + (1+\beta_2)C^2 G^2 \sum_{\tau=0}^{t-1} \beta_2^\tau \leq \frac{1+\beta_2}{1-\beta_2} C^2 G^2.
\end{aligned}
$$

Hence, using the update rule of $\tilde{v}_t$ and initialization $\tilde{v}_0 = 0$, we conclude

$$
\|\tilde{v}_t\|_{\max} \leq \max(\|v_t\|_{\max}, \|\tilde{v}_{t-1}\|_{\max}) \leq \frac{1+\beta_2}{1-\beta_2} C^2 G^2 = \frac{1+\beta_2}{1-\beta_2} \frac{(1-(1-q_r)\beta_1)^2}{(1-\beta_1)^2 (1-q_r)^2} G^2.
$$

$\square$

## E.2 NON-CONVEX ANALYSIS

**Theorem 3** (Non-convex convergence rate). *Let Assumptions 1, 2 and 3 hold. Then, choosing step-size $\eta = \min(\eta_0, \frac{1}{\sqrt{T}})$ with $\eta_0 = \frac{\epsilon}{4LC_0\sqrt{1+C_2}}$, Algorithm 3 satisfies*

$$
\begin{aligned}
\frac{1}{T}\sum_{t=1}^T \mathbb{E}[\|\nabla f(\theta_t)\|^2] &\leq 2C_0 \left(\frac{f(\theta_1) - f^*}{\sqrt{T}} + \frac{L\sigma^2}{\epsilon\sqrt{T}}\right) \\
&\quad + 4C_0 \left(\frac{f(\theta_1) - f^*}{2\eta_0 T} + \frac{L^2 C_0 C_2 \sigma^2}{2\epsilon^2 T} + \frac{(1+C_1)G^2}{\sqrt{\epsilon}T} + \frac{(1+2C_1)C_1 L G^2}{\epsilon T^{3/2}}\right),
\end{aligned}
$$

*with constants $C_0 := \sqrt{\frac{1+\beta_2}{1-\beta_2} \frac{(1-\beta_1(1-q_r))^2}{(1-\beta_1)^2(1-q_r)^2} G^2 + \epsilon}$, $C_1 := \frac{\beta_1 + (1-\beta_1)q_r}{(1-\beta_1)(1-q_r)}$, $C_2 := \frac{\beta_1 + (1-\beta_1)q_r^2}{(1-\beta_1)^2(1-q_r)^2}$.*

**Remark 1.** *Further ignoring absolute constants, the bound becomes*

$$\frac{C_0}{\sqrt{T}}\left(f(\theta_1) - f^* + \frac{L\sigma^2}{\epsilon}\right) + \frac{C_0}{T}\left(\frac{f(\theta_1) - f^*}{\eta_0} + \frac{L^2C_0C_2\sigma^2}{\epsilon^2} + \frac{C_1G^2}{\sqrt{\epsilon}} + \frac{C_1^2LG^2}{\epsilon\sqrt{T}}\right),$$

*Let us assume that for the low-rank compression of the algorithm we have $1 - q_r = \mathcal{O}(\frac{r}{d})$ (e.g., SVD option). Then $C_0 = \mathcal{O}(\frac{d}{r}G)$, $C_1 = \mathcal{O}(\frac{d}{r})$, $C_2 = \mathcal{O}(\frac{d^2}{r^2})$ and $\frac{1}{\eta_0} = \mathcal{O}(\frac{d^2}{r^2}G)$. Plugging this asymptotic expressions into the bound and ignoring other parameters (e.g., $\sigma^2, L, \epsilon$), we get*

$$\frac{d}{r}\frac{G}{\sqrt{T}} + \frac{d}{r}\frac{G}{T}\left(\frac{d^2G}{r^2} + \frac{d^3G}{r^3} + \frac{dG^2}{r} + \frac{d^2G^2}{r^2\sqrt{T}}\right)$$

*or equivalently*

$$\frac{d}{r}\frac{G}{\sqrt{T}} + \left(\frac{d}{r}\right)^4\frac{G^2}{T} + \left(\frac{d}{r}\right)^2\frac{G^3}{T} + \left(\frac{d}{r}\right)^3\frac{G^3}{T^{3/2}}.$$

*Proof.* Let $\bar{\Gamma}_t := \mathcal{U}_t\mathrm{Diag}^{-1/2}(\tilde{v}_t+\epsilon)\mathcal{U}_t^\top$ be a preconditioning matrix and $M_t := \mathcal{U}_t m_t$ the exponential moving averages in the full space. With these notations, the update rule of the model becomes $\theta_{t+1} = \theta_t - \eta_t\bar{\Gamma}_t \cdot M_t$. As it will be used later, we need to make sure that our preconditioning is positive definite. Due to the structure of $\bar{\Gamma}_t$, it is positive semi-definite only. To make it positive definite, notice that $\Gamma_t\mathcal{U}_t = \bar{\Gamma}_t\mathcal{U}_t$, where the full-rank preconditioner $\Gamma_t$ takes the form

$$\Gamma_t = \bar{\mathcal{U}}_t\mathrm{Diag}^{-1/2}(\tilde{v}_t + \epsilon, \|\tilde{v}_t\|_{\min} + \epsilon)\bar{\mathcal{U}}_t^\top, \tag{19}$$

where $\bar{\mathcal{U}}_t \in \mathbb{R}^{d\times d}$ is an orthogonal matrix with the same first $r$ columns as $\mathcal{U}_t \in \mathbb{R}^{d\times r}$ and the diagonal matrix in the middle has been extended to meet the sizes of $\bar{\mathcal{U}}_t$ by adding the values $\|\tilde{v}_t\|_{\min} + \epsilon$ on the diagonal. $\Gamma_t\mathcal{U}_t = \bar{\Gamma}_t\mathcal{U}_t$ comes from the observation that $\bar{\mathcal{U}}_t^\top\mathcal{U}_t \in \mathbb{R}^{d\times r}$ is composed of two blocks: upper $r \times r$ block of the identity matrix and $(d - r) \times r$ block of the zero matrix. Hence, the added $(d - r)$ elements on the diagonal do not really affect, so does the last $(d - r)$ columns of $\bar{\mathcal{U}}_t$. Therefore, we can write the model update rule as

$$\theta_{t+1} = \theta_t - \eta_t\Gamma_t \cdot M_t, \tag{20}$$

with full-rank preconditioning $\Gamma_t$. Next, note that

$$B_t = \beta_1 M_{t-1} + (1 - \beta_1)g_t + (1 - \beta_1)e_t, \tag{21}$$

$$m_{t-1/2} = \mathcal{U}_t^\top M_{t-1}. \tag{22}$$

Then, for the low dimensional momentum and the erorr we get

$$m_t = \beta_1 m_{t-1/2} + (1 - \beta_1)a_t = \mathcal{U}_t^\top(\beta_1 M_{t-1} + (1 - \beta_1)g_t + (1 - \beta_1)\xi_t) = \mathcal{U}_t^\top B_t, \tag{23}$$

$$(1 - \beta_1)\xi_{t+1} = (I - \mathcal{U}_t\mathcal{U}_t^\top)B_t = B_t - M_t. \tag{24}$$

Letting $B_0 = 0$, we define virtual iterates $x_t$ as follows:

$$x_{t+1} = \theta_{t+1} - \eta\Gamma_t\left((1 - \beta_1)\xi_{t+1} + \frac{\beta_1}{1 - \beta_1}B_t\right). \tag{25}$$

In particular, $x_1 = \theta_1$. Then, we derive the recurrence relation for the new sequence as follows:

$$\begin{aligned}
x_{t+1} &= \theta_{t+1} - \eta\Gamma_t\left((1 - \beta_1)\xi_{t+1} + \frac{\beta_1}{1 - \beta_1}B_t\right) \\
&\overset{(24)}{=} \theta_t - \eta\Gamma_tM_t - \eta\Gamma_t\left(B_t - M_t + \frac{\beta_1}{1 - \beta_1}B_t\right) \\
&= \theta_t - \frac{\eta}{1 - \beta_1}\Gamma_tB_t \\
&\overset{(21)}{=} \theta_t - \frac{\eta}{1 - \beta_1}\Gamma_t\left(\beta_1 M_{t-1} + (1 - \beta_1)g_t + (1 - \beta_1)\xi_t\right) \\
&\overset{(24)}{=} \theta_t - \eta\Gamma_t\left(\frac{\beta_1}{1 - \beta_1}(B_{t-1} - (1 - \beta_1)\xi_t) + \xi_t\right) - \eta\Gamma_tg_t \\
&= \theta_t - \eta\Gamma_t\left(\frac{\beta_1}{1 - \beta_1}B_{t-1} + (1 - \beta_1)\xi_t\right) - \eta\Gamma_tg_t \\
&\overset{(25)}{=} x_t - \eta\Gamma_tg_t + \eta\Delta\Gamma_t\left(\frac{\beta_1}{1 - \beta_1}B_{t-1} + (1 - \beta_1)\xi_t\right),
\end{aligned}$$

where $\Delta\Gamma_t := \Gamma_{t-1} - \Gamma_t$.

Next we apply smoothness (Assumption 1) of the loss function $f$ over the iterates $x_t$. From the gradient Lipschitzness we have

$$f(x_{t+1}) \leq f(x_t) + \langle \nabla f(x_t), x_{t+1} - x_t \rangle + \frac{L}{2} \|x_{t+1} - x_t\|^2.$$

Taking expectation, we obtain

$$
\begin{aligned}
\mathbb{E}[f(x_{t+1})] - \mathbb{E}[f(x_t)] \leq{} & -\eta \mathbb{E}\left[\langle \nabla f(x_t), \Gamma_t g_t \rangle\right] \\
& + \eta \mathbb{E}\left[\left\langle \nabla f(x_t), \Delta\Gamma_t \left(\frac{\beta_1}{1-\beta_1} B_{t-1} + (1-\beta_1)\xi_t\right) \right\rangle\right] \\
& + \frac{\eta^2 L}{2} \mathbb{E}\left[\left\|\Gamma_t g_t - \Delta\Gamma_t \left(\frac{\beta_1}{1-\beta_1} B_{t-1} + (1-\beta_1)\xi_t\right)\right\|^2\right] \\
={} & \underbrace{-\eta \mathbb{E}\left[\langle \nabla f(\theta_t), \Gamma_t g_t \rangle\right]}_{I} \tag{26} \\
& + \underbrace{\eta \mathbb{E}\left[\left\langle \nabla f(x_t), \Delta\Gamma_t \left(\frac{\beta_1}{1-\beta_1} B_{t-1} + (1-\beta_1)\xi_t\right) \right\rangle\right]}_{II} \\
& + \underbrace{\frac{\eta^2 L}{2} \mathbb{E}\left[\left\|\Gamma_t g_t - \Delta\Gamma_t \left(\frac{\beta_1}{1-\beta_1} B_{t-1} + (1-\beta_1)\xi_t\right)\right\|^2\right]}_{III} \\
& + \underbrace{\eta \mathbb{E}\left[\langle \nabla f(\theta_t) - \nabla f(x_t), \Gamma_t g_t \rangle\right]}_{IV}, \tag{27}
\end{aligned}
$$

In the following, we bound all the four terms mentioned above.

**Bounding term I.** Let $\|\Delta\Gamma\|$ be the operator norm (with respect to $\ell_2$ norm) of the matrix $\Delta\Gamma$. We have

$$
\begin{aligned}
I ={} & -\eta \mathbb{E}\left[\langle \nabla f(\theta_t), \Gamma_{t-1} g_t \rangle\right] - \eta \mathbb{E}\left[\langle \nabla f(\theta_t), \Delta\Gamma_t g_t \rangle\right] \\
\leq{} & -\eta \mathbb{E}\left[\langle \nabla f(\theta_t), \Gamma_{t-1} \nabla f(\theta_t) \rangle\right] + \eta G^2 \mathbb{E}[\|\Delta\Gamma_t\|]. \\
\leq{} & -\eta \lambda_{\min}(\Gamma_{t-1}) \mathbb{E}[\|\nabla f(\theta_t)\|^2] + \eta G^2 \mathbb{E}[\|\Delta\Gamma_t\|] \\
\leq{} & -\frac{\eta}{C_0} \mathbb{E}[\|\nabla f(\theta_t)\|^2] + \eta G^2 \mathbb{E}[\|\Delta\Gamma_t\|], \tag{28}
\end{aligned}
$$

where we use Assumption 2 and Lemma 4 to bound

$$\lambda_{\min}(\Gamma_{t-1}) \geq (\|\tilde{v}_{t-1}\|_{\max} + \epsilon)^{-1/2} \geq \left(\frac{1+\beta_2}{1-\beta_2} \frac{(1-\beta_1(1-q_r))^2}{(1-\beta_1)^2(1-q_r)^2} G^2 + \epsilon\right)^{-1/2} = \frac{1}{C_0}.$$

Note that the purpose of making $\Gamma$ matrix positive definite is to have negative term in (28).

**Bounding term II.** Then we have

$$
\begin{aligned}
II \leq{} & \eta \mathbb{E}\left[\left\langle \nabla f(\theta_t), \Delta\Gamma_t \left(\frac{\beta_1}{1-\beta_1} B_{t-1} + (1-\beta_1)\xi_t\right) \right\rangle\right] \\
& + \eta \mathbb{E}\left[\left\langle \nabla f(x_t) - \nabla f(\theta_t), \Delta\Gamma_t \left(\frac{\beta_1}{1-\beta_1} B_{t-1} + (1-\beta_1)\xi_t\right) \right\rangle\right] \\
\leq{} & \eta \mathbb{E}\left[\|\nabla f(\theta_t)\| \left\|\Delta\Gamma_t \left(\frac{\beta_1}{1-\beta_1} B_{t-1} + (1-\beta_1)\xi_t\right)\right\|\right] \\
& + \eta^2 L \mathbb{E}\left[\left\|\Gamma_t \left(\frac{\beta_1}{1-\beta_1} B_{t-1} + (1-\beta_1)\xi_t\right)\right\| \cdot \left\|\Delta\Gamma_t \left(\frac{\beta_1}{1-\beta_1} B_{t-1} + (1-\beta_1)\xi_t\right)\right\|\right] \\
\overset{(15)}{\leq}{} & \eta C_1 G^2 \mathbb{E}[\|\Delta\Gamma_t\|] + \frac{\eta^2 C_1^2 L G^2}{\sqrt{\epsilon}} \mathbb{E}[\|\Delta\Gamma_t\|], \tag{29}
\end{aligned}
$$

where $C_1 = \frac{\beta_1 + (1-\beta_1)q_r}{(1-\beta_1)(1-q_r)}$ and we used the fact that the largest eigenvalue $\lambda_{\max}(\Gamma_t) = \|\Gamma_t\| = (\|\tilde{v}_t\|_{\min} + \epsilon)^{-1/2} \leq \epsilon^{-1/2}$. The second inequality is because of smoothness of $f(\theta)$, and the last inequality is due to Lemma 1, Assumption 2 and the property of norms.

**Bounding term III.** This term can be bounded as follows:

$$
\begin{aligned}
III &\leq \eta^2 L \mathbb{E}\left[\|\Gamma_t g_t\|^2\right] + \eta^2 L \mathbb{E}\left[\left\|\Delta\Gamma_t \left(\frac{\beta_1}{1-\beta_1}B_{t-1} + (1-\beta_1)\xi_t\right)\right\|^2\right] \\
&\leq \frac{\eta^2 L}{\epsilon}\mathbb{E}[\|g_t - \nabla f(\theta_t) + \nabla f(\theta_t)\|^2] + \eta^2 L \mathbb{E}\left[\left\|\Delta\Gamma_t \left(\frac{\beta_1}{1-\beta_1}B_{t-1} + (1-\beta_1)\xi_t\right)\right\|^2\right] \\
&\leq \frac{\eta^2 L}{\epsilon}\left(\mathbb{E}[\|\nabla f(\theta_t)\|^2] + \sigma^2\right) + \eta^2 C_1^2 LG^2 \mathbb{E}[\|\Delta\Gamma_t\|^2] \\
&\leq \frac{\eta^2 L}{\epsilon}\mathbb{E}[\|\nabla f(\theta_t)\|^2] + \frac{\eta^2 L\sigma^2}{\epsilon} + \eta^2 C_1^2 LG^2 \mathbb{E}[\|\Delta\Gamma_t\|^2],
\end{aligned}
\tag{30}
$$

where we used Assumption 3 that $g_t$ is unbiased with bounded variance $\sigma^2$.

**Bounding term IV.** We have

$$
\begin{aligned}
IV &= \eta\mathbb{E}\left[\langle\nabla f(\theta_t) - \nabla f(x_t), \Gamma_{t-1}g_t\rangle\right] + \eta\mathbb{E}\left[\langle\nabla f(\theta_t) - \nabla f(x_t), \Delta\Gamma_t g_t\rangle\right] \\
&\leq \eta\mathbb{E}\left[\langle\nabla f(\theta_t) - \nabla f(x_t), \Gamma_{t-1}\nabla f(\theta_t)\rangle\right] + \eta^2 L\mathbb{E}\left[\left\|\Gamma_t\left(\frac{\beta_1}{1-\beta_1}B_{t-1} + (1-\beta_1)\xi_t\right)\right\|\|\Delta\Gamma_t g_t\|\right] \\
&\overset{(a)}{\leq} \frac{\eta\rho}{2\epsilon}\mathbb{E}[\|\nabla f(\theta_t)\|^2] + \frac{\eta}{2\rho}\mathbb{E}[\|\nabla f(\theta_t) - \nabla f(x_t)\|^2] + \frac{\eta^2 C_1 LG^2}{\sqrt{\epsilon}}\mathbb{E}[\|\Delta\Gamma_t\|] \\
&\overset{(b)}{\leq} \frac{\eta\rho}{2\epsilon}\mathbb{E}[\|\nabla f(\theta_t)\|^2] + \frac{\eta^3 L^2}{2\rho}\mathbb{E}\left[\left\|\Gamma_t\left(\frac{\beta_1}{1-\beta_1}B_{t-1} + (1-\beta_1)\xi_t\right)\right\|^2\right] + \frac{\eta^2 C_1 LG^2}{\sqrt{\epsilon}}\mathbb{E}[\|\Delta\Gamma_t\|] \\
&\leq \frac{\eta\rho}{2\epsilon}\mathbb{E}[\|\nabla f(\theta_t)\|^2] + \frac{\eta^3 L^2}{2\rho\epsilon}\mathbb{E}\left[\left\|\frac{\beta_1}{1-\beta_1}B_{t-1} + (1-\beta_1)\xi_t\right\|^2\right] + \frac{\eta^2 LC_1 G^2}{\sqrt{\epsilon}}\mathbb{E}[\|\Delta\Gamma_t\|],
\end{aligned}
\tag{31}
$$

where (a) is due to Young's inequality and (b) is based on Assumption 1. Now integrating (28), (29), (30), (31) into (27),

$$
\begin{aligned}
I &\leq -\eta\lambda_{\min}(\Gamma_{t-1})\mathbb{E}[\|\nabla f(\theta_t)\|^2] + \eta G^2 \mathbb{E}[\|\Delta\Gamma_t\|] \\
II &\leq \eta C_1 G^2 \mathbb{E}[\|\Delta\Gamma_t\|] + \frac{\eta^2 C_1^2 LG^2}{\sqrt{\epsilon}}\mathbb{E}[\|\Delta\Gamma_t\|] \\
III &\leq \frac{\eta^2 L}{\epsilon}\mathbb{E}[\|\nabla f(\theta_t)\|^2] + \frac{\eta^2 L\sigma^2}{\epsilon} + \eta^2 C_1^2 LG^2 \mathbb{E}[\|\Delta\Gamma_t\|^2] \\
IV &\leq \frac{\eta\rho}{2\epsilon}\mathbb{E}[\|\nabla f(\theta_t)\|^2] + \frac{\eta^3 L^2}{2\rho\epsilon}\mathbb{E}\left[\left\|\frac{\beta_1}{1-\beta_1}B_{t-1} + (1-\beta_1)\xi_t\right\|^2\right] + \frac{\eta^2 LC_1 G^2}{\sqrt{\epsilon}}\mathbb{E}[\|\Delta\Gamma_t\|],
\end{aligned}
$$

and taking the telescoping summation over $t = 1, \ldots, T$, we obtain

$$\mathbb{E}[f(x_{T+1}) - f(x_1)]$$

$$\leq \left( -\frac{\eta}{C_0} + \frac{\eta^2 L}{\epsilon} + \frac{\eta\rho}{2\epsilon} \right) \sum_{t=1}^{T} \mathbb{E}[\|\nabla f(\theta_t)\|^2] + \frac{T\eta^2 L\sigma^2}{\epsilon} + \frac{\eta^3 L^2}{2\rho\epsilon} \sum_{t=1}^{T} \mathbb{E}\left[ \left\| \frac{\beta_1}{1-\beta_1} B_{t-1} + (1-\beta_1)\xi_t \right\|^2 \right]$$

$$+ \left( \eta(1+C_1)G^2 + \frac{\eta^2(1+C_1)C_1 LG^2}{\sqrt{\epsilon}} \right) \sum_{t=1}^{T} \mathbb{E}[\|\Delta\Gamma_t\|] + \eta^2 C_1^2 LG^2 \sum_{t=1}^{T} \mathbb{E}[\|\Delta\Gamma_t\|^2]$$

$$\overset{(16)}{\leq} \left( -\frac{\eta}{C_0} + \frac{\eta^2 L}{\epsilon} + \frac{\eta\rho}{2\epsilon} + \frac{\eta^3 L^2 C_2}{2\rho\epsilon} \right) \sum_{t=1}^{T} \mathbb{E}[\|\nabla f(\theta_t)\|^2] + \frac{T\eta^2 L\sigma^2}{\epsilon} + \frac{T\eta^3 L^2 C_2 \sigma^2}{2\rho\epsilon}$$

$$+ \left( \eta(1+C_1)G^2 + \frac{\eta^2(1+C_1)C_1 LG^2}{\sqrt{\epsilon}} \right) \sum_{t=1}^{T} \mathbb{E}[\|\Delta\Gamma_t\|] + \eta^2 C_1^2 LG^2 \sum_{t=1}^{T} \mathbb{E}[\|\Delta\Gamma_t\|^2],$$

where we used (16) of Lemma 1 with constant $C_2 = \frac{\beta_1 + (1-\beta_1)q_r^2}{(1-\beta_1)^2(1-q_r)^2}$. Choosing $\rho = \frac{\epsilon}{2C_0}$ and $\eta \leq \eta_0 := \frac{\epsilon}{4LC_0\sqrt{1+C_2}}$ and using Lemma 3, we get

$$\mathbb{E}[f(x_{T+1}) - f(x_1)] \leq -\frac{\eta}{2C_0} \sum_{t=1}^{T} \mathbb{E}[\|\nabla f(\theta_t)\|^2] + \frac{T\eta^2 L\sigma^2}{\epsilon} + \frac{T\eta^3 L^2 C_0 C_2 \sigma^2}{\epsilon^2}$$

$$+ \frac{2\eta(1+C_1)G^2}{\sqrt{\epsilon}} + \frac{2\eta^2(1+2C_1)C_1 LG^2}{\epsilon}.$$

Re-arranging terms, we get that

$$\frac{1}{T} \sum_{t=1}^{T} \mathbb{E}[\|\nabla f(\theta_t)\|^2] \leq 2C_0 \left( \frac{f(\theta_1) - f^*}{T\eta} + \frac{\eta L\sigma^2}{\epsilon} + \frac{\eta^2 L^2 C_0 C_2 \sigma^2}{\epsilon^2} \right)$$

$$+ 4C_0 \left( \frac{(1+C_1)G^2}{T\sqrt{\epsilon}} + \frac{\eta(1+2C_1)C_1 LG^2}{T\epsilon} \right),$$

where in the last inequality we used $x_1 = \theta_1$ and the lower bound $f^* \leq f(\theta)$ for all $\theta \in \mathbb{R}^d$. Finally, choosing $\eta = \min(\eta_0, \frac{1}{\sqrt{T}})$ and considering the two cases, we arrive at the following rate

$$\frac{1}{T} \sum_{t=1}^{T} \mathbb{E}[\|\nabla f(\theta_t)\|^2] \leq 2C_0 \left( \max\left(1, \frac{1}{\eta_0\sqrt{T}}\right) \frac{f(\theta_1) - f^*}{\sqrt{T}} + \frac{L\sigma^2}{\epsilon\sqrt{T}} + \frac{L^2 C_0 C_2 \sigma^2}{\epsilon^2 T} \right)$$

$$+ 4C_0 \left( \frac{(1+C_1)G^2}{\sqrt{\epsilon}T} + \frac{(1+2C_1)C_1 LG^2}{\epsilon T^{3/2}} \right)$$

$$\leq 2C_0 \left( \frac{f(\theta_1) - f^*}{\sqrt{T}} + \frac{L\sigma^2}{\epsilon\sqrt{T}} \right)$$

$$+ 4C_0 \left( \frac{f(\theta_1) - f^*}{2\eta_0 T} + \frac{L^2 C_0 C_2 \sigma^2}{2\epsilon^2 T} + \frac{(1+C_1)G^2}{\sqrt{\epsilon}T} + \frac{(1+2C_1)C_1 LG^2}{\epsilon T^{3/2}} \right),$$

which completes the proof of the theorem. $\qquad\square$

### E.3 ANALYSIS UNDER PL CONDITION

As in the non-convex analysis, here we derive the convergence rate with fixed step-size $\eta$.

**Theorem 4. (Convergence rate under PL)** *Let Assumptions 1, 2, 3 and 4 hold. Then, choosing step-size $\eta = \min(\eta_0, \frac{2C_0 \log T}{\mu T})$ with $\eta_0 = \min(\frac{\epsilon}{16LC_0}, \frac{C_0(1-\beta_1)(1-q_r)}{2\mu}, \frac{\epsilon^{3/4}}{6L\sqrt{C_0 C_2}})$, Algorithm 3 satisfies*

$$\mathbb{E}[f(\theta_{T+1})] - f^* \leq \frac{\log T}{T} \left( \frac{2LC_0^2 \sigma^2}{\mu^2\epsilon} + \frac{6C_0(1+C_1)G^2}{\mu\sqrt{\epsilon}} \right) + \widetilde{\mathcal{O}}\left( \frac{G^4}{T^2} \right),$$

*Proof.* We start from descent lemma

$$
\mathbb{E}[f(x_{t+1})] - \mathbb{E}[f(x_t)] = \underbrace{-\eta\mathbb{E}\left[\langle \nabla f(x_t), \Gamma_t g_t \rangle\right]}_{I'}
$$

$$
+ \underbrace{\eta\mathbb{E}\left[\left\langle \nabla f(x_t), \Delta\Gamma_t\left(\frac{\beta_1}{1-\beta_1}B_{t-1} + (1-\beta_1\xi_t)\right)\right\rangle\right]}_{II}
$$

$$
+ \underbrace{\frac{\eta^2 L}{2}\mathbb{E}\left[\left\|\Gamma_t g_t - \Delta\Gamma_t\left(\frac{\beta_1}{1-\beta_1}B_{t-1} + (1-\beta_1\xi_t)\right)\right\|^2\right]}_{III}. \quad (32)
$$

We bound part $II$ and part $III$ in the same way as it was done in the non-convex analysis. We now provide a bound for part $I'$:

$$
\begin{aligned}
I' =& -\eta\mathbb{E}\left[\langle \nabla f(x_t), \Gamma_t g_t \rangle\right]\\
=& -\eta\mathbb{E}\left[\langle \nabla f(x_t), \Gamma_{t-1} g_t \rangle\right] - \eta\mathbb{E}\left[\langle \nabla f(x_t), \Delta\Gamma_t g_t \rangle\right]\\
=& -\eta\mathbb{E}\left[\langle \nabla f(x_t), \Gamma_{t-1}(g_t - \nabla f(x_t) + \nabla f(x_t)) \rangle\right] - \eta\mathbb{E}\left[\langle \nabla f(x_t), \Delta\Gamma_t g_t \rangle\right]\\
=& -\eta\mathbb{E}\left[\langle \nabla f(x_t), \Gamma_{t-1}\nabla f(x_t) \rangle\right] - \eta\mathbb{E}\left[\langle \nabla f(x_t), \Gamma_{t-1}(g_t - \nabla f(x_t)) \rangle\right]\\
& -\eta\mathbb{E}\left[\langle \nabla f(x_t), \Delta\Gamma_t g_t \rangle\right].
\end{aligned}
$$

We further expand and bound this equation as follows:

$$
\begin{aligned}
I' \leq& -\frac{\eta}{C_0}\mathbb{E}\left[\|\nabla f(x_t)\|^2\right]\\
& -\eta\mathbb{E}\left[\langle \nabla f(x_t) - \nabla f(\theta_t) + \nabla f(\theta_t), \Gamma_{t-1}(\nabla f(\theta_t) - \nabla f(x_t)) \rangle\right]\\
& -\eta\mathbb{E}\left[\langle \nabla f(x_t) - \nabla f(\theta_t) + \nabla f(\theta_t), \Delta\Gamma_t g_t \rangle\right]\\
=& -\frac{\eta}{C_0}\mathbb{E}\left[\|\nabla f(x_t)\|^2\right]\\
& -\eta\mathbb{E}\left[\langle \nabla f(x_t) - \nabla f(\theta_t), \Gamma_{t-1}(\nabla f(\theta_t) - \nabla f(x_t)) \rangle\right]\\
& -\eta\mathbb{E}\left[\langle \nabla f(\theta_t), \Gamma_{t-1}(\nabla f(\theta_t) - \nabla f(x_t)) \rangle\right]\\
& -\eta\mathbb{E}\left[\langle \nabla f(x_t) - \nabla f(\theta_t), \Delta\Gamma_t g_t \rangle\right]\\
& -\eta\mathbb{E}\left[\langle \nabla f(\theta_t), \Delta\Gamma_t g_t \rangle\right]\\
\leq& -\frac{\eta}{C_0}\mathbb{E}\left[\|\nabla f(x_t)\|^2\right] + \frac{\eta}{\sqrt{\epsilon}}\mathbb{E}\left[\|\nabla f(x_t) - f(\theta_t)\|^2\right]\\
& -\eta\mathbb{E}\left[\langle \nabla f(\theta_t), \Gamma_{t-1}(\nabla f(\theta_t) - \nabla f(x_t)) \rangle\right]\\
& +\eta\mathbb{E}\left[\langle \nabla f(x_t) - \nabla f(\theta_t), \Delta\Gamma_t g_t \rangle\right] + \eta\mathbb{E}\left[\langle \nabla f(\theta_t), \Delta\Gamma_t g_t \rangle\right].
\end{aligned}
$$

To bound the second and third terms above we reuse derivation done in (31) to have

$$
\eta\mathbb{E}\left[\langle \nabla f(\theta_t), \Gamma_{t-1}(\nabla f(\theta_t) - \nabla f(x_t)) \rangle\right] \overset{(31)}{\leq} \frac{\eta\rho}{2\epsilon}\mathbb{E}[\|\nabla f(\theta_t)\|^2] \quad (33)
$$

$$
+ \frac{\eta^3 L^2}{2\rho\epsilon}\mathbb{E}\left[\left\|\frac{\beta_1}{1-\beta_1}B_{t-1} + (1-\beta_1)\xi_t\right\|^2\right]
$$

$$
\mathbb{E}\left[\|\nabla f(x_t) - f(\theta_t)\|^2\right] \overset{(31)}{\leq} \frac{\eta^2 L^2}{\epsilon}\mathbb{E}\left[\left\|\frac{\beta_1}{1-\beta_1}B_{t-1} + (1-\beta_1)\xi_t\right\|^2\right] \quad (34)
$$

Next, we use the Cauchy–Schwartz inequality to bound inner products above, $L$-smoothness inequality to bound $\|\nabla f(x_t) - \nabla f(\theta_t)\| \leq L\|x_t - \theta_t\| \leq \frac{\eta L C_1 G}{\sqrt{\epsilon}}$, and the inequality $-\|a\|^2 \leq$

$-\frac{1}{2}\|b\|^2 + \|a-b\|^2$ for the first term:

$$
\begin{aligned}
I' \leq & -\frac{\eta}{C_0}\mathbb{E}\left[\|\nabla f(x_t)\|^2\right] + \frac{\eta}{\sqrt{\epsilon}}\mathbb{E}\left[\|\nabla f(x_t) - f(\theta_t)\|^2\right] \\
& + \frac{\eta\rho}{2\epsilon}\mathbb{E}[\|\nabla f(\theta_t)\|^2] + \frac{\eta^3 L^2}{2\rho\epsilon}\mathbb{E}\left[\left\|\frac{\beta_1}{1-\beta_1}B_{t-1} + (1-\beta_1)\xi_t\right\|^2\right] \\
& + \eta G\mathbb{E}\left[\|\nabla f(x_t) - \nabla f(\theta_t)\|\|\Delta\Gamma_t\|\right] + \eta G^2\mathbb{E}\left[\|\Delta\Gamma_t\|\right] \\
\leq & -\frac{\eta}{2C_0}\mathbb{E}\left[\|\nabla f(x_t)\|^2\right] - \frac{\eta}{2C_0}\mathbb{E}\left[\|\nabla f(x_t)\|^2\right] \\
& + \frac{\eta\rho}{2\epsilon}\mathbb{E}[\|\nabla f(\theta_t)\|^2] + \frac{\eta}{\sqrt{\epsilon}}\mathbb{E}\left[\|\nabla f(x_t) - f(\theta_t)\|^2\right] + \frac{\eta^3 L^2}{2\rho\epsilon}\mathbb{E}\left[\left\|\frac{\beta_1}{1-\beta_1}B_{t-1} + (1-\beta_1)\xi_t\right\|^2\right] \\
& + \eta G\frac{\eta L C_1 G}{\sqrt{\epsilon}}\mathbb{E}\left[\|\Delta\Gamma_t\|\right] + \eta G^2\mathbb{E}\left[\|\Delta\Gamma_t\|\right] \\
\leq & -\frac{\eta}{2C_0}\mathbb{E}\left[\|\nabla f(x_t)\|^2\right] - \frac{\eta}{4C_0}\mathbb{E}[\|\nabla f(\theta_t)\|^2] + \frac{\eta}{2C_0}\mathbb{E}[\|\nabla f(x_t) - \nabla f(\theta_t)\|^2] \\
& + \frac{\eta\rho}{2\epsilon}\mathbb{E}[\|\nabla f(\theta_t)\|^2] + \frac{\eta}{\sqrt{\epsilon}}\mathbb{E}\left[\|\nabla f(x_t) - f(\theta_t)\|^2\right] + \frac{\eta^3 L^2}{2\rho\epsilon}\mathbb{E}\left[\left\|\frac{\beta_1}{1-\beta_1}B_{t-1} + (1-\beta_1)\xi_t\right\|^2\right] \\
& + \frac{\eta^2 L C_1 G^2}{\sqrt{\epsilon}}\mathbb{E}\left[\|\Delta\Gamma_t\|\right] + \eta G^2\mathbb{E}\left[\|\Delta\Gamma_t\|\right] \\
\leq & -\frac{\eta}{2C_0}\mathbb{E}\left[\|\nabla f(x_t)\|^2\right] - \frac{\eta}{4C_0}\mathbb{E}[\|\nabla f(\theta_t)\|^2] \\
& + \frac{\eta\rho}{2\epsilon}\mathbb{E}[\|\nabla f(\theta_t)\|^2] + \frac{3\eta}{2\sqrt{\epsilon}}\mathbb{E}\left[\|\nabla f(x_t) - f(\theta_t)\|^2\right] + \frac{\eta^3 L^2}{2\rho\epsilon}\mathbb{E}\left[\left\|\frac{\beta_1}{1-\beta_1}B_{t-1} + (1-\beta_1)\xi_t\right\|^2\right] \\
& + \frac{\eta^2 L C_1 G^2}{\sqrt{\epsilon}}\mathbb{E}\left[\|\Delta\Gamma_t\|\right] + \eta G^2\mathbb{E}\left[\|\Delta\Gamma_t\|\right] \\
\overset{(34)}{\leq} & -\frac{\eta}{2C_0}\mathbb{E}\left[\|\nabla f(x_t)\|^2\right] - \frac{\eta}{4C_0}\mathbb{E}[\|\nabla f(\theta_t)\|^2] + \frac{\eta\rho}{2\epsilon}\mathbb{E}[\|\nabla f(\theta_t)\|^2] \\
& + \frac{3\eta}{2\sqrt{\epsilon}}\frac{\eta^2 L^2}{\epsilon}\mathbb{E}\left[\left\|\frac{\beta_1}{1-\beta_1}B_{t-1} + (1-\beta_1)\xi_t\right\|^2\right] + \frac{\eta^3 L^2}{2\rho\epsilon}\mathbb{E}\left[\left\|\frac{\beta_1}{1-\beta_1}B_{t-1} + (1-\beta_1)\xi_t\right\|^2\right] \\
& + \frac{\eta^2 L C_1 G^2}{\sqrt{\epsilon}}\mathbb{E}\left[\|\Delta\Gamma_t\|\right] + \eta G^2\mathbb{E}\left[\|\Delta\Gamma_t\|\right] \\
\leq & -\frac{\eta}{2C_0}\mathbb{E}\left[\|\nabla f(x_t)\|^2\right] - \frac{\eta}{4C_0}\mathbb{E}[\|\nabla f(\theta_t)\|^2] + \frac{\eta\rho}{2\epsilon}\mathbb{E}[\|\nabla f(\theta_t)\|^2] \\
& + \frac{\eta^3 L^2}{2\epsilon}\left(\frac{3}{\sqrt{\epsilon}} + \frac{1}{\rho}\right)\mathbb{E}\left[\left\|\frac{\beta_1}{1-\beta_1}B_{t-1} + (1-\beta_1)\xi_t\right\|^2\right] + \frac{\eta^2 L C_1 G^2}{\sqrt{\epsilon}}\mathbb{E}\left[\|\Delta\Gamma_t\|\right] + \eta G^2\mathbb{E}\left[\|\Delta\Gamma_t\|\right].
\end{aligned}
$$

Plugging the obtained bound for $I'$ with previously obtained bounds for $II$ and $III$

$$
\begin{aligned}
II & \leq \eta C_1 G^2\mathbb{E}[\|\Delta\Gamma_t\|] + \frac{\eta^2 C_1^2 L G^2}{\sqrt{\epsilon}}\mathbb{E}[\|\Delta\Gamma_t\|] \\
III & \leq \frac{\eta^2 L}{\epsilon}\mathbb{E}[\|\nabla f(\theta_t)\|^2] + \frac{\eta^2 L\sigma^2}{\epsilon} + \eta^2 C_1^2 L G^2\mathbb{E}[\|\Delta\Gamma_t\|^2]
\end{aligned}
$$

into (32), choosing $\rho = \frac{\epsilon}{8C_0}$ and using the step-size bound $\eta \leq \frac{\epsilon}{16LC_0}$ we get

$$
\begin{aligned}
\mathbb{E}[f(x_{t+1})] - \mathbb{E}[f(x_t)] \leq & -\frac{\eta}{2C_0}\mathbb{E}\left[\|\nabla f(x_t)\|^2\right] - \frac{\eta}{4C_0}\mathbb{E}[\|\nabla f(\theta_t)\|^2] + \frac{\eta\rho}{2\epsilon}\mathbb{E}[\|\nabla f(\theta_t)\|^2] \\
& + \frac{\eta^3 L^2}{2\epsilon}\left(\frac{3}{\sqrt{\epsilon}} + \frac{1}{\rho}\right)\mathbb{E}\left[\left\|\frac{\beta_1}{1-\beta_1}B_{t-1} + (1-\beta_1)\xi_t\right\|^2\right] + \frac{\eta^2 LC_1 G^2}{\sqrt{\epsilon}}\mathbb{E}\left[\|\Delta\Gamma_t\|\right] \\
& + \eta C_1 G^2 \mathbb{E}[\|\Delta\Gamma_t\|] + \frac{\eta^2 C_1^2 LG^2}{\sqrt{\epsilon}}\mathbb{E}[\|\Delta\Gamma_t\|] + \eta G^2 \mathbb{E}\left[\|\Delta\Gamma_t\|\right] \\
& + \frac{\eta^2 L}{\epsilon}\mathbb{E}[\|\nabla f(\theta_t)\|^2] + \frac{\eta^2 L\sigma^2}{\epsilon} + \eta^2 C_1^2 LG^2 \mathbb{E}[\|\Delta\Gamma_t\|^2] \\
\leq & -\frac{\eta}{2C_0}\mathbb{E}\left[\|\nabla f(x_t)\|^2\right] - \frac{\eta}{8C_0}\mathbb{E}[\|\nabla f(\theta_t)\|^2] + \frac{\eta^2 L\sigma^2}{\epsilon} \\
& + \frac{\eta^3 L^2}{2\epsilon}\left(\frac{3}{\sqrt{\epsilon}} + \frac{\epsilon}{8C_0}\right)\mathbb{E}\left[\left\|\frac{\beta_1}{1-\beta_1}B_{t-1} + (1-\beta_1)\xi_t\right\|^2\right] \\
& + \eta(1+C_1)G^2\mathbb{E}[\|\Delta\Gamma_t\|] + \frac{\eta^2(1+C_1)C_1 LG^2}{\sqrt{\epsilon}}\mathbb{E}[\|\Delta\Gamma_t\|] + \eta^2 C_1^2 LG^2\mathbb{E}[\|\Delta\Gamma_t\|^2] \\
\leq & -\frac{\eta\mu}{C_0}(\mathbb{E}[f(x_t)] - f^*) - \frac{\eta}{8C_0}\mathbb{E}[\|\nabla f(\theta_t)\|^2] + \frac{\eta^2 L\sigma^2}{\epsilon} \\
& + \frac{2\eta^3 L^2}{\epsilon^{3/2}}\mathbb{E}\left[\left\|\frac{\beta_1}{1-\beta_1}B_{t-1} + (1-\beta_1)\xi_t\right\|^2\right] \\
& + \eta(1+C_1)G^2\mathbb{E}[\|\Delta\Gamma_t\|] + \frac{\eta^2(1+C_1)C_1 LG^2}{\sqrt{\epsilon}}\mathbb{E}[\|\Delta\Gamma_t\|] + \eta^2 C_1^2 LG^2\mathbb{E}[\|\Delta\Gamma_t\|^2],
\end{aligned}
$$

where in the last inequality we applied PL condition from Assumption 4. After some reshuffling of the terms, we obtain the following recursion:

$$
\begin{aligned}
\mathbb{E}[f(x_{t+1})] - f^* \leq & \left(1 - \frac{\eta\mu}{C_0}\right)(\mathbb{E}[f(x_t)] - f^*) - \frac{\eta}{8C_0}\mathbb{E}[\|\nabla f(\theta_t)\|^2] + \frac{\eta^2 L\sigma^2}{\epsilon} \\
& + \frac{2\eta^3 L^2}{\epsilon^{3/2}}\mathbb{E}\left[\left\|\frac{\beta_1}{1-\beta_1}B_{t-1} + (1-\beta_1)\xi_t\right\|^2\right] \\
& + \eta(1+C_1)G^2\mathbb{E}[\|\Delta\Gamma_t\|] + \frac{\eta^2(1+C_1)C_1 LG^2}{\sqrt{\epsilon}}\mathbb{E}[\|\Delta\Gamma_t\|] + \eta^2 C_1^2 LG^2\mathbb{E}[\|\Delta\Gamma_t\|^2].
\end{aligned}
$$

Now we unroll the obtained recursion and invoke Lemma 2 with $\gamma = 1 - \frac{\eta\mu}{C_0}$. From $\eta \leq \frac{\epsilon}{4LC_0} \leq \frac{C_0}{4\mu}$, we conclude that $\gamma = 1 - \frac{\eta\mu}{C_0} \in (0,1)$. To satisfy the condition on $\gamma$ of Lemma 1, we enforce the

bound $\eta \leq \frac{C_0}{2\mu}(1 - \beta_1)(1 - q_r)$ on the step size. Therefore,

$$
\mathbb{E}[f(x_{T+1})] - f^* \overset{(16)}{\leq} \left(1 - \frac{\eta\mu}{C_0}\right)^T (f(x_1) - f^*) - \frac{\eta}{8C_0} \sum_{t=1}^{T} \left(1 - \frac{\eta\mu}{C_0}\right)^{T-t} \mathbb{E}[\|\nabla f(\theta_t)\|^2]
$$
$$
+ \frac{\eta^2 L\sigma^2}{\epsilon} \sum_{t=0}^{T-1} \left(1 - \frac{\eta\mu}{C_0}\right)^t
$$
$$
+ \frac{2\eta^3 L^2}{\epsilon^{3/2}} \sum_{t=1}^{T} \left(1 - \frac{\eta\mu}{C_0}\right)^{T-t} \mathbb{E}\left[\left\|\frac{\beta_1}{1 - \beta_1} B_{t-1} + (1 - \beta_1)\xi_t\right\|^2\right]
$$
$$
+ \eta(1 + C_1)G^2 \sum_{t=1}^{T} \left(1 - \frac{\eta\mu}{C_0}\right)^{T-t} \mathbb{E}[\|\Delta\Gamma_t\|]
$$
$$
+ \frac{\eta^2(1 + C_1)C_1 LG^2}{\sqrt{\epsilon}} \sum_{t=1}^{T} \left(1 - \frac{\eta\mu}{C_0}\right)^{T-t} \mathbb{E}[\|\Delta\Gamma_t\|]
$$
$$
+ \eta^2 C_1^2 LG^2 \sum_{t=1}^{T} \left(1 - \frac{\eta\mu}{C_0}\right)^{T-t} \mathbb{E}[\|\Delta\Gamma_t\|^2]. \tag{35}
$$

For the second sum above we upper bound it by its infinite sum as

$$
\sum_{t=0}^{T-1} \left(1 - \frac{\eta\mu}{C_0}\right)^t \leq \sum_{t=0}^{\infty} \left(1 - \frac{\eta\mu}{C_0}\right)^t = \frac{C_0}{\eta\mu}.
$$

For the third sum, we apply Lemma 2 and for the other three sums we bound $1 - \frac{\eta\mu}{C_0} \leq 1$ and apply the bounds in Lemma 3:

$$
\sum_{t=1}^{T} \left(1 - \frac{\eta\mu}{C_0}\right)^t \mathbb{E}[\|\Delta\Gamma_t\|] \leq \sum_{t=1}^{T} \mathbb{E}[\|\Delta\Gamma_t\|] \leq \frac{2}{\sqrt{\epsilon}},
$$

$$
\sum_{t=1}^{T} \left(1 - \frac{\eta\mu}{C_0}\right)^t \mathbb{E}[\|\Delta\Gamma_t\|^2] \leq \sum_{t=1}^{T} \mathbb{E}[\|\Delta\Gamma_t\|^2] \leq \frac{2}{\epsilon}.
$$

Plugging all this bounds into (35), cancelling terms involving gradients with step size restriction $\eta \le \frac{\epsilon^{3/4}}{6L\sqrt{C_0 C_2}}$ and noticing that $x_1 = \theta_1$, we finally get

$$
\begin{aligned}
\mathbb{E}[f(x_{T+1})] - f^* \le{} & \left(1 - \frac{\eta\mu}{C_0}\right)^T (f(\theta_1) - f^*) - \frac{\eta}{8C_0} \sum_{t=1}^{T} \left(1 - \frac{\eta\mu}{C_0}\right)^{T-t} \mathbb{E}[\|\nabla f(\theta_t)\|^2] \\
& + \frac{\eta^2 L\sigma^2}{\epsilon} \frac{C_0}{\eta\mu} \\
& + \frac{2\eta^3 L^2}{\epsilon^{3/2}} \cdot 2C_2 \left(\frac{C_0\sigma^2}{\eta\mu} + \sum_{t=1}^{T} \left(1 - \frac{\eta\mu}{C_0}\right)^{T-t} \mathbb{E}[\|\nabla f(\theta_t)\|^2]\right) \\
& + \eta(1+C_1)G^2 \frac{2}{\sqrt{\epsilon}} + \frac{\eta^2(1+C_1)C_1 LG^2}{\sqrt{\epsilon}} \frac{2}{\sqrt{\epsilon}} + \eta^2 C_1^2 LG^2 \frac{2}{\epsilon} \\
\le{} & \left(1 - \frac{\eta\mu}{C_0}\right)^T (f(\theta_1) - f^*) \\
& + \frac{\eta L C_0\sigma^2}{\mu\epsilon} + \frac{4\eta^2 L^2 C_0 C_2 \sigma^2}{\mu\epsilon^{3/2}} \\
& + \frac{2\eta(1+C_1)G^2}{\sqrt{\epsilon}} + \frac{2\eta^2(1+C_1)C_1 LG^2}{\epsilon} + \frac{2\eta^2 C_1^2 LG^2}{\epsilon} \\
={} & \left(1 - \frac{\eta\mu}{C_0}\right)^T (f(\theta_1) - f^*) \\
& + \eta\left(\frac{L C_0\sigma^2}{\mu\epsilon} + \frac{2(1+C_1)G^2}{\sqrt{\epsilon}}\right) + \eta^2\left(\frac{4L^2 C_0 C_2\sigma^2}{\mu\epsilon^{3/2}} + \frac{2(1+2C_1)C_1 LG^2}{\epsilon}\right).
\end{aligned}
$$

The obtained rate above is with respect to the virtual iterates $x_t$ that we defined for the purposes of analysis. To convert this rate with respect to the iterates $\theta_t$ of the algorithm, we apply $L$-smoothness to bound the functional difference:

$$
|f(x_t) - f(\theta_t)| \le |\langle \nabla f(\theta_t), x_t - \theta_t\rangle| + \frac{L}{2}\|x_t - \theta_t\|^2 \le \frac{\eta C_1 G^2}{\sqrt{\epsilon}} + \frac{\eta^2 L C_1^2 G^2}{2\epsilon},
$$

which implies

$$
\begin{aligned}
\mathbb{E}[f(\theta_{T+1})] - f^* \le{} & \left(1 - \frac{\eta\mu}{C_0}\right)^T (f(\theta_1) - f^*) \\
& + \eta\left(\frac{L C_0\sigma^2}{\mu\epsilon} + \frac{3(1+C_1)G^2}{\sqrt{\epsilon}}\right) + \eta^2\left(\frac{4L^2 C_0 C_2\sigma^2}{\mu\epsilon^{3/2}} + \frac{5(1+C_1)C_1 LG^2}{\epsilon}\right).
\end{aligned}
$$

Plugging $\eta = \min(\eta_0, \frac{2C_0 \log T}{\mu T})$ with $\eta_0 = \min(\frac{\epsilon}{16 L C_0}, \frac{C_0(1-\beta_1)(1-q_r)}{2\mu}, \frac{\epsilon^{3/4}}{6L\sqrt{C_0 C_2}})$, we get

$$
\begin{aligned}
\mathbb{E}[f(\theta_{T+1})] - f^* \le{} & \max\left(\frac{1}{T^2}, \left(1 - \frac{\eta_0\mu}{C_0}\right)^T\right)(f(\theta_1) - f^*) \\
& + \frac{\log T}{T} \frac{2C_0}{\mu}\left(\frac{L C_0\sigma^2}{\mu\epsilon} + \frac{3(1+C_1)G^2}{\sqrt{\epsilon}}\right) \\
& + \frac{\log^2 T}{T^2} \frac{4C_0^2}{\mu^2}\left(\frac{4L^2 C_0 C_2\sigma^2}{\mu\epsilon^{3/2}} + \frac{5(1+C_1)C_1 LG^2}{\epsilon}\right) \\
={} & \frac{\log T}{T}\left(\frac{2L C_0^2\sigma^2}{\mu^2\epsilon} + \frac{6C_0(1+C_1)G^2}{\mu\sqrt{\epsilon}}\right) + \tilde{\mathcal{O}}\left(\frac{G^4}{T^2}\right),
\end{aligned}
$$

which completes the proof. $\qquad\square$

