# OpenReview forum: "LDAdam: Adaptive Optimization from Low-Dimensional Gradient Statistics"
_ICLR.cc/2025/Conference — ICLR 2025 Poster_

### Official Review · Reviewer_tZ3Y · 2024-10-30

**Soundness:** 3
**Presentation:** 3
**Contribution:** 3
**Rating:** 8
**Confidence:** 4

**Summary:**

Thin paper introduces a novel memory-efficient low-dimensional Adam optimizer for training large models. Specifically, the authors propose a new projection-aware update for the optimizer states that allows for transitioning between subspaces. They also perform block power iteration to at each step to efficiently and accurately project the gradient and optimizer states. To reduce the error caused by the low-rank projection, this work leverages a new generalized error feedback mechanism that accounts for both gradient and optimizer state compression. The authors theoretically analyze the convergence rates for different objectives. To validate the proposed algorithm, the authors present extensive experimental results on different datasets with the comparison to different baselines.

**Strengths:**

1. The investigated topic is quite interesting and critical, particularly when LLMs are popular in every area. A memory-efficient optimizer is required to conduct computationally efficient fine-tuning.
2. The paper is easy to follow and the presentation in this work is clear.
3. This work presents in-depth theoretical analysis for the proposed algorithm and show the explicits the convergence rate.
4. The empirical evidence to validate the proposed algorithm look extensive and thorough.
5. The comparison between the proposed method and baselines is technically convincing.

**Weaknesses:**

1. Assumption 2 is a quite strong assumption for the gradient, which may limit the applicability of the proposed algorithm. Until now, most existing algorithms have relaxed this assumption in their works such that this assumption imposes some weaknesses to the proposed algorithm.
2. In this work, the authors have mentioned that GaLore is the closest work. Though we have seen the authors have shown in detail how the proposed algorithm differs, it would be great to include a technical discussion for the difference between them in terms of convergence rate. That way, it is clearer to directly show the exact theoretical difference.
3. In the practical view of the algorithm, the authors use Gram-Schmidt process to calculate $\mathcal{U}_t$. However, the process has numerical instability, computational intensity, and even inaccuracies. If this process fails, how can the authors guarantee that there always exists a feasible and reliable $\mathcal{U}_t$ for the subsequent steps?
4. Although the experimental results look promising, how to validate the theoretical conclusions from Theorem 1 and Theorem 2, particularly, the impact of the compression ratio? The authors need to present some ablation studies for the proposed algorithm.


**********************Post-rebuttal Comment***************************
After carefully reading the rebuttal from the authors and taking other reviewers' comments into consideration, I think the authors have addressed my major concerns. The new results also look interesting to me. Thanks for the efforts. I will also raise my score accordingly.

**Questions:**

1. How to relax the bounded gradient assumption in the study?
2. How to guarantee a feasible $\mathcal{U}_t$ if Gram-Schmidt process fails?
3. How to validate the theoretical conclusions from both theorems?

---

> ### Author Response · Authors · 2024-11-20
> **Authors' Answers #1 for Reviewer tZ3Y**
>
> **1. About bounded gradient assumption**
>
> While we appreciate the reviewer’s feedback on our analysis approach, we wish to gently push back on the statement questioning our use of the bounded gradient assumption by emphasizing that we are in the adaptive optimization setting.
>
> Specifically, we believe that your statement, “most existing algorithms have relaxed this assumption in their works,” does not apply for the analysis of *adaptive optimization algorithms like Adam*. In fact, to our knowledge, most (if not all) existing analyses for Adam-type optimizers use either bounded gradient or bounded variance assumptions in some form or another.
>
> We believe that the reviewer may be referring to relaxing this assumption in the analysis of SGD-type optimizers: in this case, the bounded gradient assumption can indeed be removed. In our analysis, we can also remove the bounded gradient condition if we drop the adaptive preconditioning in LDAdam and consider its SGD-type base. However, our clear aim is to design an Adam-type optimizer for deep learning applications.
>
> As we mentioned in the main text, this bounded gradient assumption is standard in the adaptive optimization literature: please see (Reddi et al., 2019; Chen et al., 2019; Defossez et al., 2022; Li et al., 2022; Xie et al., 2023; Zhou et al., 2024; Modoranu et al., 2024). Attempting to relax the bounded gradient condition, some works (Shi et al., 2020; Zhang et al., 2022; Wang et al., 2022) resorted to using a strong growth condition (which is not necessarily a relaxation over bounded gradient condition) or bounded stochastic noise conditions (Li et al., 2023) (Hong and Lin, 2024). Extremely recent work–appearing just a couple of weeks ago–achieved an optimal rate by modifying Adam and only slightly relaxing the bounded gradient assumption by requiring the *expected* gradient norm to be bounded (Taniguchi et al., NeurIPS 2024).
>
> We hope this justifies our use of this standard assumption–we will further clarify this point in the next version of the paper.
>
> **2. Difference between LDAdam and GaLore**
>
> In our view, the analytical assumptions made in GaLore are quite non-standard, and therefore it is impossible to perform a like-for-like comparison of the analyses.
> Concretely, please note that the theoretical justification and convergence analysis in Galore refers to a vanilla SGD optimizer (please see Lemma 3.3 and Theorem 3.8 there), and uses *fixed* low-rank projection matrices throughout the optimization process, which is not something that the actual algorithm implements. Importantly, for the analysis to work, it is also assumed that the minimal eigenvalues of some matrices need to be extremely large: please see the min_t kappa_t lower bound.
> By contrast, our analysis is performed under standard preconditions for adaptive optimization: we make essentially the same assumptions as the analysis of AMSGrad / AMSComp. Thus, our work unifies the practical direction of research into memory-efficient optimization, initiated by GaLore, with the large body of work on analyzing various efficient versions of Adam/AMSGrad under a standard set of assumptions.
>
> **References:**
>
> (Shi et al., 2020) Naichen Shi, Dawei Li, Mingyi Hong, and Ruoyu Sun. Rmsprop converges with proper hyperparameter. In International Conference on Learning Representations, 2020.
>
> (Zhang et al., 2022) Yushun Zhang, Congliang Chen, Naichen Shi, Ruoyu Sun, and Zhi-Quan Luo. Adam can converge without any modification on update rules. Advances in Neural Information Processing Systems, 2022.
>
> (Wang et al., 2022) Bohan Wang, Yushun Zhang, Huishuai Zhang, Qi Meng, Zhi-Ming Ma, Tie-Yan Liu, and Wei Chen. Provable adaptivity in adam. arXiv preprint arXiv:2208.09900, 2022.
>
> (Li et al., 2023) Haochuan Li, Ali Jadbabaie, and Alexander Rakhlin. Convergence of Adam under relaxed assumptions. arXiv preprint arXiv:2304.13972, 2023.
>
> (Hong and Lin, 2024) Yusu Hong, Junhong Lin. On Convergence of Adam for Stochastic Optimization under Relaxed Assumptions. Conference on Neural Information Processing Systems, 2024.
>
> (Taniguchi et al., 2024) Shohei Taniguchi, Keno Harada, Gouki Minegishi, Yuta Oshima, Seong Cheol Jeong, Go Nagahara, Tomoshi Iiyama, Masahiro Suzuki, Yusuke Iwasawa, Yutaka Matsuo. ADOPT: Modified Adam Can Converge with Any β2 with the Optimal Rate. Neural Information Processing Systems, 2024.

---

> ### Author Response · Authors · 2024-11-20
> **Authors' Answers #2 for Reviewer tZ3Y**
>
> **3. About Gram-Schmidt Process**
>
> The prerequisite for our method is to compute the singular value decomposition (SVD) of the full-rank gradient. Since computing the SVD at each step is prohibitively expensive, we decided to use block power iteration with warm start [1], followed by orthogonalization of the resulting set of vectors. We represent the orthogonalization step as the execution of the Gram-Schmidt process. However, the Gram-Schmidt process is equivalent to applying the QR decomposition and keeping only the orthonormal basis provided by Q. In practice, we rely on the PyTorch implementation to compute the QR decomposition. (The latter is more stable than a custom implementation of the Gram-Schmidt process.)
>
> To your point, for degenerate or singular matrices, the Q matrices may not be full rank (i.e., the dimension of $span(Q)$ is less than $r$). In these extreme cases, power iteration will propagate such a rank deficiency and harm the optimization process. In practice, we have not observed any failure of the learning subspace adaptation process, and the method has already been shown to be robust by the PowerSGD method [1], even at large scale [3].
> Thus, we have not yet implemented a specific mechanism to correct such a deficiency. However, one could easily counteract such extreme situations by restarting the power iteration process for a new computation of the full rank singular value decomposition.
> Finally, one can use any other computationally efficient method to compute the gradient SVD. In particular, we also tried the Torch implementation of the low-rank SVD method [2]. However, this did not make any noticeable difference, and we preferred to revert to the more transparent and well-established strategy used in PowerSGD [1].
>
> On a more technical note, we would like to mention that the Block Power Iteration we are using for LDAdam is different from the one used in Power SGD. Both methods estimate the first $r$ eigenvectors, but in different ways. For example, Power SGD does not use QR decomposition, but it performs Gram Schmidt column by column, while we use a QR decomposition.
>
> **4. About theoretical conclusions (impact of compression ratio)**
>
> To validate the theoretical conclusions on the impact of the compression ratio, we pretrained Llama 350M with increasing ranks. We added to the appendix section (figure 4) training dynamics and validation perplexity with respect to the rank.
>
> To allow for comparison between training dynamics, we reported training dynamics with the same learning rate. We observe faster convergence with respect to the rank, which is in accordance with the theoretical results stating that the convergence depends on a slowdown factor ($d/r$ for non convex loss (theorem 1)  and $(d/r)^2$ for non convex loss under PL assumption (theorem 2)) that is decreasing in $r$.
>
> We further show that the best validation perplexity over the learning rage set {5e-4, 1e-3, 5e-3} is decreasing in the rank $r$. In particular, interestingly, we observe that the decrease in validation perplexity is roughly logarithmic with respect to the rank.
>
> **References:**
>
> [1] Thijs Vogels, Sai Praneeth Karimireddy, and Martin Jaggi. PowerSGD: Practical Low-Rank Gradient Compression for Distributed Optimization. arXiv preprint arXiv:1905.13727, 2020.
>
> [2] Nathan Halko, Per-Gunnar Martinsson, and Joel A. Tropp. Finding Structure with Randomness: Probabilistic Algorithms for Constructing Approximate Matrix Decompositions. arXiv preprint arXiv:0909.4061, 2010.
>
> [3] https://medium.com/pytorch/accelerating-pytorch-ddp-by-10x-with-powersgd-585aef12881d

---

> > ### Comment · Reviewer_tZ3Y · 2024-11-22
> >
> > 4. About theoretical conclusions (impact of compression ratio)
> >
> > Thanks for adding the figure to showcase the impact of compression ratio. The results look interesting. I think the theoretical results have now been somewhat validated.

---

> > > ### Author Response · Authors · 2024-11-25
> > > **Thank you!**
> > >
> > > Thank you, we are glad you found our results interesting!

---

> ### Comment · Reviewer_tZ3Y · 2024-11-22
> **Response to rebuttal**
>
> 1. About bounded gradient assumption
>
> Thanks so much for your clarification. Your points are taken and make sense to me. What I meant was if you could try to even relax the assumption for adaptive optimization in your analysis, probably by coming up with new techniques. But I understand this could be out of the scope of this study.

---

### Official Review · Reviewer_2kSc · 2024-11-02

**Soundness:** 2
**Presentation:** 3
**Contribution:** 2
**Rating:** 6
**Confidence:** 3

**Summary:**

This paper on LDAdam addresses a crucial challenge in training large models: reducing memory consumption of optimizers without sacrificing model quality. The key innovation is the introduction of a lower-dimensional subspace optimization approach, where adaptive optimization occurs within constrained dimensions. The optimizer, though operating in a subspace, ensures exploration across the entire parameter space, thereby maintaining model expressivity while being memory efficient.

**Strengths:**

1. **Projection-Aware Update Rule**: LDAdam uses a projection-aware update rule to transition between subspaces. This rule allows it to estimate the gradient statistics even after dimensional reduction, adapting efficiently to changes in the subspace basis without losing essential gradient information.

2. **Generalized Error Feedback Mechanism**: To address inaccuracies from low-rank projections, LDAdam introduces a unique error feedback mechanism that accounts for both gradient and optimizer state compression. This mechanism reintroduces projection errors in subsequent iterations, ensuring stability in updates and helping LDAdam match the performance of uncompressed optimizers.

3. **Convergence and Memory Efficiency**: The paper provides a proof of LDAdam’s convergence under standard assumptions. With this design, LDAdam achieves a significantly reduced memory footprint. In practical terms, LDAdam’s optimizer states require only a fraction of the memory needed by Adam, making it feasible to train large language models in resource-constrained environments.

**Weaknesses:**

Liang et al. [1] provide a convergence analysis for GaLore and related algorithms without relying on a 'stable-rank' assumption. Since LDAdam aligns with this framework, I recommend citing [1] for its convergence insights. Additionally, a comparison of LDAdam’s convergence analysis with [1], specifically on the impact of removing the 'stable-rank' assumption, would help clarify the authors’ theoretical contributions.

[1] Memory-Efficient LLM Training with Online Subspace Descent, Kaizhao Liang, Bo Liu, Lizhang Chen, Qiang Liu.

**Questions:**

The original GaLore paper supports gradient accumulation, contrary to the authors' claim. I suggest they clarify this point, either correcting the claim or explaining any differences in their interpretation or implementation of GaLore to avoid potential misunderstandings.

Good paper, I suggest acceptance.

---

> ### Author Response · Authors · 2024-11-20
> **Authors' Answers for Reviewer 2kSc**
>
> **1. About stable-rank assumption**
>
> Thank you for bringing up the work [1], which we will include in our revision. In brief, this work studies the convergence of optimizers by examining their continuous-time ODE forms in the limit of infinitesimal step size. The paper shows that adding a low-rank projection matrix to an optimizer following the Hamiltonian descent structure does not change the Lyapunov function. Hence, the dynamically projected optimizer obeys similar convergence analysis as long as the degenerate cases from projection are excluded.
>
> Similar to [1], we also exclude degenerate cases from projections by enforcing the condition q_r<1. However, the other parts of the analyses and approaches are fundamentally different. In contrast to our analysis and results, convergence analysis in [1] uses infinitesimal step size, does not reveal any convergence rate, does not consider stochastic noise coming from mini-batching, and assumes that the solution of the Hamiltonian system is bounded.
>
> Generally, providing a convergence rate for a discrete-time optimizer is much more challenging than mere convergence of its continuous-time analog. One notable challenge is properly integrating an error feedback mechanism in the presence of compression, which is necessary to guarantee general worst-case theoretical convergence rates. Our theoretical analysis follows the line of work of (Zhou et al., 2024) (Reddi et al., 2019) (Li et al., 2022) (He et al., 2023) and (Modoranu et al., 2024), to mention a few.
>
> **2. About gradient accumulation**
>
> In brief, please note that GaLore can support gradient accumulation, but in standard frameworks like Pytorch this would lead to memory overheads that are proportional to the model size, and would therefore cancel out some of the key benefits of GaLore.
>
> This point is not made very clear in the original GaLore paper. The original code for the GaLore algorithm is slightly confusing, because the training loop provided does accept  hyperparameters for gradient accumulation.
>
> However, looking deeper into the implementation, we observe that GaLore’s algorithm performs per-layer weight updates, which rely on gradient hook and is incompatible with gradient accumulation. This is clearly stated in the PyTorch documentation : “How to save memory by fusing the optimizer step into the backward pass (https://pytorch.org/tutorials/intermediate/optimizer_step_in_backward_tutorial.html). Specifically, for gradient low-rank projection to support gradient accumulation, *one must retain the gradient*, which would lead to a memory overhead equal to the model size.
>
> This is what we meant by our original remark. To clarify this point and address your concern, we updated the optimizer comparison Table 1, and added a detailed explanation to our comment regarding the GaLore algorithm, which can be found in Appendix D.
>
> Please let us know if this clarifies the situation; we would be happy to clarify this further if necessary!
>
> **References:**
>
> [1] Memory-Efficient LLM Training with Online Subspace Descent, Kaizhao Liang, Bo Liu, Lizhang Chen, Qiang Liu.

---

> > ### Author Response · Authors · 2024-11-25
> > **Discussion reminder**
> >
> > Dear Reviewer,
> >
> > As the discussion period is almost over, we wanted to send you a very gentle reminder to examine our response and let us know if it addressed your concerns.
> >
> > With best regards,\
> > The submission authors

---

> ### Comment · Reviewer_2kSc · 2024-12-02
>
> Dear Authors,
>
> Thank you for your detailed responses and revisions. I am satisfied that my concerns have been fully addressed, and your updates have strengthened the paper. I will increase my score accordingly.
>
> I do have one minor observation: you mentioned that "the convergence analysis in [1] uses infinitesimal step size and does not reveal any convergence rate." However, since [1] provides a construction of a monotonic Hamiltonian, discretizing the continuous-time system should be straightforward and could yield a convergence rate with minimal effort. Similarly, for the stochastic setting, adding a variance term, as shown in [2], could address this effectively.
>
> [1] Memory-Efficient LLM Training with Online Subspace Descent, Kaizhao Liang, Bo Liu, Lizhang Chen, Qiang Liu.
>
> [2] Momentum Ensures Convergence of SIGNSGD under Weaker Assumptions, Tao Sun, Qingsong Wang, Dongsheng Li, Bao Wang.
>
> Best regards,
>
> 2kSc

---

### Official Review · Reviewer_HAQR · 2024-11-03

**Soundness:** 3
**Presentation:** 3
**Contribution:** 3
**Rating:** 8
**Confidence:** 3

**Summary:**

The paper proposes a new memory-efficient training algorithm based on Adam via efficient projection onto subspaces. The proposed method enjoys non-asymptotic theoretical rate of convergence and demonstrate good experimental results in training large language models (LLMs).

**Strengths:**

The paper is well-written and the contribution is clear. The algorithm is novel with theoretical convergence. The experiments indicates that the new approach is performing not only better than GaLore, but also better than the vanilla Adam for some parameter settings.

**Weaknesses:**

(Please respond to the "Questions" directly) The computational time of proposed method is not clear; Some discussion on the theoretical results is needed.

**Questions:**

Overall I think this is a work with interesting observations and methods. I have the following concerns and questions (I'm happy to increase my evaluation if the authors address the issues):

1. To my understanding, GaLore doesn't employ SVD step for every iteration in order to save computation overheads. On the contrary, the proposed method would need a Gram-Schmidt process and several extra steps for updating the momentums and gradients. A major conern is the computation overhead of this proposed method. I udnerstand that the memory consumption is roughly the same as GaLore, yet I'm curious about the time required for training LLMs both in theory and in all the experiment settings using the proposed method.
2. The convergence results in Theorem 1 and 2 are similar to that of full-rank Adagrad/Adam. To my understanding, the low rank parameter $r$ only plays a role in the constant $q_r$? If so, can the author comment on how to choose such $r$ and if the theory has any implication on the choice of $r$. In particular, if $r=m$ and the method reduce back to the original Adam, can we say any matching result with the analysis on the original Adam?
3. This question is related to the previous one. The experiments show that LDAdam is even better than original Adam in some cases (Table 3 with $r=512$, Table 4 with $r=256$). Can the authors expand on the discussion of "regularization effect of compression" or provide some references? It's still not very clear to me right now.
4. Any experiment on even larger LLMs such as Llama 1b or 7b? I'm curious what will happen if the model is large and the training is conducted in parallel on multiple GPUs.

---

> ### Author Response · Authors · 2024-11-20
> **Authors' Answers for Reviewer HAQR**
>
> **1. About computation overhead**
>
> We agree that computational complexity is a key concern, and LDAdam’s implementation is carefully designed to avoid overheads. To address this more clearly, we provided convergence graphs in Perplexity-vs-running-time format (Figure 2), as well as ablations of Throughput (tokens/second) and Peak memory usage versus rank (Figure 3). We have also added results on a 1.3B-scale model.
>
> The timing results (Figure 2) show that LDAdam does not add overheads in terms of perplexity reached versus time: specifically, the slightly better convergence of LDAdam relative to Adam compensates for the computational overheads of the projections, which are in the order of 10% per step. We highlight that this occurs at both 350M and 1.3B parameter scales.  Moreover, Figure 3 shows that the memory and runtime trade-offs scale smoothly with rank in our implementation.
>
> The figures we refer to can be found in the PDF at the following anonymous link: https://anonymous.4open.science/r/LDAdam-anonymous-1583/4387_LDAdam_rebuttal_figures.pdf
>
> **2. About the role of $r$ in the constant $q_r$**
>
> Your observation that low-rank parameter r only affects the constant $q_r$ is accurate. In particular, the lower the rank r is, the larger the value of $q_r<1$ becomes. As the constant appears in convergence rates by the expression $1/(1 - q_r$), lower rank implies roughly m/r times more optimization steps and bigger computational overhead (see the discussions after the theorems). So, from the perspective of computational overhead, the theory suggests choosing a rank as high as possible. However, if we focus on the memory overhead, then theory favors lower ranks as we would reduce the optimizer states' memory by the same factor of m/r times. Thus, which overhead is more critical will decide how to choose the rank.
>
> In practice, a simple “greedy” method for choosing the rank is picking the maximum value supported by the available hardware resources (i.e., the max rank that fits in GPU/CPU memory). The key message of our theory is that regardless of the selected rank (as long as the constant $q_r < 1$), the proposed LDAdam method has asymptotically the same theoretical convergence guarantees as the full-rank version, although the constants will indeed change depending on the chosen rank.
>
> Regarding the matching result with full-rank Adam, as mentioned in the discussion paragraphs after the theorems, the asymptotic rates align with the rate of uncompressed AMSGrad (a provably-convergent version of Adam) in the stochastic non-convex setup (Zhou et al., 2024) or in the PL setup (He et al., 2023).
>
> **3. About the regularization effect of compression**
>
> We were careful to avoid claiming that our optimizer is “better” than Adam, since this is an extremely versatile, robust and practically-successful optimizer. However, we acknowledge that a significant fraction of our experiments suggest that LDAdam can lead to slight increases in accuracy, for both fine-tuning and training from scratch.
>
> While we do not have a full explanation for this phenomenon, our findings are in agreement with a related line of work that investigates the effect of regularization in optimization, and shows that while explicit regularization is a technique to improve the generalization behavior of overparameterized deep neural networks, the gradient descent-based optimization process leads to implicit regularization [1, 2]. Our intuition is that low-rank updates, which are equivalent to sparse updates in an adapted coordinate system, favor such regularization relative to full-rank Adam.
>
> **References:**
>
> [1] Chiyuan Zhang, Samy Bengio, Moritz Hardt, Benjamin Recht, and Oriol Vinyals. Understanding Deep Learning Requires Rethinking Generalization. arXiv preprint arXiv:1611.03530, 2017.
>
> [2] Behnam Neyshabur. Implicit Regularization in Deep Learning. arXiv preprint arXiv:1709.01953, 2017.

---

> > ### Comment · Reviewer_HAQR · 2024-11-23
> >
> > I thank the author for the rebuttal. In general I think the new results are interesting and I encourage the authors to incorporate them in the paper, especially the new plot w.r.t. training time, and the table summarizing the optimizers (both of them should be in the main paper content). I'd like to increase my evaluation score.

---

> > > ### Author Response · Authors · 2024-11-25
> > > **Thank you!**
> > >
> > > Many thanks for your response, we are very glad you found our response convincing! \
> > > We will of course incorporate the new results into the paper.

---

### Official Review · Reviewer_hwuW · 2024-11-05

**Soundness:** 3
**Presentation:** 3
**Contribution:** 3
**Rating:** 6
**Confidence:** 2

**Summary:**

The paper proposes a new optimization algorithm to train large (language) models.
Like the recent GaLore algorithm, the proposed algorithm is a variation of ADAM, that performs updates in a lower-dimensional space, to store lower-dimensional quantities to be more memory-efficient.
As I understand (and as written by the authors), the real meat behind this kind of algorithm is the choice of the projection to obtain a meaningful lower-dimensional subspace that is both efficient to compute/store and a good enough approximation of the gradient. The proposed algorithm retains the desirable memory-efficient properties of GaLore, while almost matching the generalization performance of ADAM. Convergence proofs and exhaustive experiments on text benchmarks are provided.

**Strengths:**

The paper is overall easy to follow, the idea is interesting, and seems to empirically works great.

**Weaknesses:**

I see that the authors provide some running times and memory metrics in Appendix B, but it seems to me that this part should be more exhaustive in order to prove the proposed algorithm useful.
I know this is hard to monitor for algorithms on GPU, but could authors provide graphs similar to Figure 1, but with running times? and memory usage? with ADAM, GaLore, and LDADAM (proposed).
In other words, could authors provide graphs with perplexity as a function of runtime and perplexity as a function of memory usage? Even if these required graphs will be imperfect for a lot reasons, I think they are crucial to gain insights and potentially prove usefulness.

**Questions:**

see Weaknesses

---

> ### Author Response · Authors · 2024-11-20
> **Authors' Answers for Reviewer hwuW**
>
> > the performance part should be more exhaustive
>
> We acknowledge this point. To address it, we have made three additions to the revision, which we outline here:
>
> 1. We have added a perplexity-vs-running time graph, in Figure 2. This also now contains a training run on a 1.3B model.
> Specifically, Figure 2 shows that, for both 350M and 1.3B parameter count, LDAdam presents a favorable trade-off in terms of loss-versus-running time, when compared to Adam.
>
> 2. We have added a more detailed discussion of the asymptotic (theoretical) costs of the various algorithms, instantiating runtimes to the methods we use. This can be found in Table 5 and the surrounding text.
>
> 3. To address the trade-off between memory usage and runtime, we present, in Figure 3, a detailed ablation of Tokens/second vs rank used, and Peak memory vs rank used.
> This validates our theoretical analysis, and shows that the overheads of LDAdam scale smoothly with respect to rank.
> We hope this fully addresses your concern, and would be happy to follow up if necessary.
>
> The figures we refer to can be found in the PDF at the following anonymous link: https://anonymous.4open.science/r/LDAdam-anonymous-1583/4387_LDAdam_rebuttal_figures.pdf

---

### Author Response · Authors · 2024-11-20
**Authors' General Response for the Rebuttal**

We would like to thank the reviewers for their useful and diverse feedback! We provide individual answers to all questions, as well as additional experimental results, as asked for by the reviewers. Specifically, we outline the main questions addressed by our rebuttal and included in our PDF revision:

- To address questions by Reviewers, we have added pretraining experiments with up to 1.3B parameter models, and expanded our algorithm to the multi-GPU setting. The results show the same trends as in the original submission, where LDAdam has slightly better convergence than Adam, with a much lower memory overhead. They also confirm the scalability and stability of our approach.

- We have added an in-depth discussion of computational cost, with detailed asymptotics for LDAdam and competing approaches. This shows that LDAdam provides the same or better memory cost relative to prior work on memory-efficient optimization, while at the same time providing better convergence guarantees. We have also provided results in convergence-vs-wall-clock-time format as well as ablations between throughput (tokens/second) and rank, outlining the smooth performance trade-offs that LDAdam’s implementation allows.

- We have also added several detailed answers, such as clarifying that the bounded gradient assumption is standard in the analysis of adaptive optimization methods, discussing the superiority of our analysis approach relative to that of GaLore.

- We discussed the influence of the rank parameter r on computation and memory overhead, particularly in the context of practical considerations for selecting its value. As a sanity check, we reconfirmed that using the maximum possible rank allows our theory to match the results of the original Adam/AMSGrad. Additionally, we provided a detailed technical comparison between our theoretical framework and results and those of GaLore and approaches based on continuous-time ODEs. Finally, we justified our use of the bounded gradient assumption, which is a standard practice in the adaptive optimization literature, as there is no strictly better alternative.

---

> ### Author Response · Authors · 2024-11-22
> **Discussion reminder**
>
> Dear reviewers,
>
> As the discussion period will be drawing to a close soon, we wanted to ask if you could please examine our responses and let us know if they addressed your concerns.
>
> With best regards,\
> The authors

---

### Comment · Area_Chair_Nm2u · 2024-11-25
**Important: Please Review Rebuttals and Update Reviews as Needed**

Dear Reviewers,

Thank you for your hard work and dedication to providing thoughtful reviews for this year’s ICLR submissions. Your efforts play a vital role in maintaining the conference’s high standards and fostering meaningful discussions in the community.

As we are close to the end of the discussion phase, I kindly urge you to read the authors’ responses and reevaluate your reviews carefully, especially if they address your concerns. If you find that the authors have resolved the issues or clarified misunderstandings, please consider adjusting your comments and scores accordingly. This ensures fairness and gives each submission the opportunity to be judged on its merits.

Your continued commitment is greatly appreciated—thank you for contributing to the success of ICLR!

---

### Meta-Review · Area_Chair_Nm2u · 2024-12-18

**Metareview:**

This paper presents LDAdam, a memory-efficient optimizer for training large models. The main idea is to perform adaptive optimization in low-dimensional subspaces while still exploring the full parameter space. The paper’s strengths are in its theoretical guarantees and practical performance. The projection-aware update rule helps to transition between subspaces smoothly, keeping the gradient statistics accurate even with dimensional reduction. The generalized error feedback mechanism helps reduce errors from low-rank projections, which allows LDAdam to perform similarly or better compared to full-rank Adam. The authors give detailed convergence proofs and show strong experiments, including results for models up to 1.3 billion parameters. The results show LDAdam is scalable, stable, and uses much less memory.

Reviewers found the paper clear and the idea interesting. The experiments show that LDAdam saves memory and performs better than baseline optimizers like Adam and GaLore in some cases. The authors answered reviewers’ concerns about computational cost, convergence rates, and compression ratios with more experiments, such as perplexity vs. runtime graphs and throughput vs. rank studies. Some concerns about bounded gradient assumptions and the Gram-Schmidt process were also addressed well. While there are some minor issues, the reviewers agree the paper is a good contribution to memory-efficient optimization methods.

**Additional Comments On Reviewer Discussion:**

The discussion among reviewers was helpful and led to a clearer understanding of the paper. The authors provided detailed responses that addressed most of the concerns, especially regarding the computational overhead, convergence rates, and the impact of low-rank compression. New results, like the graphs showing perplexity vs. runtime and the analysis of memory usage, helped to support the claims in the paper. Reviewers seemed satisfied with these clarifications.

---

### Decision · Program_Chairs · 2025-01-22

Accept (Poster)